# Functional Integration with Process Mining and Process Analyzing for Structural and Behavioral Properness Validation of Processes Discovered from Event Log Datasets

**Kwanghoon Pio Kim**

Data and Process Engineering Research Lab., Division of Computer Science and Engineering,
Kyonggi University, Gyeonggi-do 16227, Korea; kwang@kgu.ac.kr; Tel.: +82-10-2059-8522

**Abstract:** In this paper, we propose an integrated approach for seamlessly and effectively providing the mining and the analyzing functionalities to redesigning work for very large-scale and massively parallel process models that are discovered from their enactment event logs. The integrated approach especially aims at analyzing not only their structural complexity and correctness but also their animation-based behavioral properness, and becomes concretized to a sophisticated analyzer. The core function of the analyzer is to discover a very large-scale and massively parallel process model from a process log dataset and to validate the structural complexity and the syntactical and behavioral properness of the discovered process model. Finally, this paper writes up the detailed description of the system architecture with its functional integration of process mining and process analyzing. More precisely, we excogitate a series of functional algorithms for extracting the structural constructs and for visualizing the behavioral properness of those discovered very large-scale and massively parallel process models. As experimental validation, we apply the proposed approach and analyzer to a couple of process enactment event log datasets available on the website of the 4TU.Centre for Research Data.

**Keywords:** structured information control net; process mining; process analyzing; structural analysis; behavioral analysis; process rediscovery

---

## 1. Introduction

A (business) process management system (BPMS) is defined as a system that fully automates the definition, analysis, deployment, execution, and monitoring and controlling of work procedures in a process-aware enterprise. One of the essential components of BPMS is the modeling subsystem (Buildtime) that is substantially supported by a graphical and formal methodology of the process model. The conventional modeling subsystem is equipped with a series of functional components supporting everything from the modeling of a process model with graphical notations to analyzing and deploying it onto the enacting subsystem (runtime). In recent times, keeping abreast of the process mining [1,2] with the emerging concept of enterprise big data, the advanced BPMSs have begun to be furnished with the process mining functionality [3] as an essential subsystem (Miningtime). In other words, the time of process reengineering and redesigning has truly come in process-aware enterprises and organizations. This atmosphere booming process redesigning and reengineering is becoming a catalyst for triggering the emergence of the concepts of process mining and process-centered knowledge mining that rediscover several perspectives of processes, such as control flow, data flow [4], resource allocation planning [5], social and organizational perspectives from their execution histories, and traces collected at runtime. In consequence, the time of functional integration of the rediscovery functionality

of the Miningtime subsystem and the analysis functionality of the Buildtime subsystem ought to be coming. Actually, the motivation and necessity of this functional integration came up with a couple of experimental validation studies of the process mining algorithm. Note that one of the minded process models is made up of 624 business activities and so it is almost impossible to fulfill any further meaningful analytical work due to the huge number of activities.

The purpose of this paper is to integrate the verification functionality (process analyzing) of the Buildtime subsystem with the rediscovery functionality (process mining) of the Miningtime subsystem, and through which the Buildtime subsystem is seamlessly charged with visual verification and structural analysis [6–8] of the process models rediscovered from their enactment event trace datasets by the Miningtime subsystem. In general, the process-aware enterprises' operational goal ought to be minimize the loss of effectiveness and the depreciation of efficiency in controlling the life-cycles of the deployed process models. Thus, the process designer wants to inspect the syntactical correctness as well as the structural performance of the corresponding process models prior to redesigning and reengineering them. These inspection activities might be much more burdensome if the rediscovered process models are especially characterized with very large-scale and massively parallel structures [9]. At this moment, it is necessary to remind that the Miningtime subsystem's functionality is to rediscover a process model from its enactment event trace dataset and to represent the rediscovered one via both the textual and graphical representation language and notation like XPDL [10] and BPMN [11], respectively. (XPDL stands for XML process definition language and is the standardized XML specification released by the workflow management coalition; BPMN stands for business process modeling notation and is the standardized graphical notation released by the BMIDTF (business modeling integration domain task force) of the object management group.) Again, the Miningtime subsystem eventually transforms the graphical model into the textual model represented by anyone of the standardized process definition languages. Getting back to the point, and thus if we want to minimize the loss of effectiveness and the depreciation of efficiency in managing the life-cycle [12] of the rediscovered process model, it is very important to support a certain fashion of automated means in inspecting its syntactical correctness as well as its structural properness, before deploying the rediscovered process model.

Therefore, the most important benefit of the proposed approach ought to be the seamless integration of the process mining functionality and the process analyzing functionality. Once we discover a process model, then we need to analyze various aspects of the discovered process model. Especially, this paper concerns the cases of when the discovered models are very large-scale and massively parallel in terms of their structural building blocks. In particular, we assume that those process models are represented as structured information control nets and are rediscovered from their enactment event trace datasets by the process mining and analyzing system. That is, in order to efficiently and seamlessly verify the structural correctness (matched pairing and proper nesting) [13,14] of the discovered very large-scale and massively parallel process models, we develop a process mining and analyzing system based upon the functional integration approach proposed in this paper. The implemented system supports the structural verification and the visual simulation [15] with analytical control-path animation. The structural components of rediscovered process model are made up of the entity types, such as activity, role, actor, invoked applications, and relevant data, and the association types, such as control-flow association, data-flow association, actor-role association, activity-role association, and activity-application association. In other words, the system not only applies the structural verification functionality to the associative components of a discovered process model, but also applies the behavioral simulation and animation functionality to the entity components of the rediscovered process model. We assume that the rediscovered process model is textually represented in the standardized XML-based process definition language (XPDL) [10], and that it is also graphically represented in a form of the structured information control nets (SICN) [14,16].

In terms of organizing the paper, we describe a functional architecture integrating the process analyzing and the process mining right after the literature surveys in the next section. In the

ensuing sections, we specify the formal and graphical structural characteristics of the very large-scale and massively parallel process models, describe the functional details and design artifacts of the architectural components, and describe carrying out a couple of experimental deployments and analyses for those very large-scale and massively parallel process models, which were discovered from a dataset that was released from the 4TU.Centre for Research Data [17].

## 2. Related Works

This paper aims to integrate the process mining functionality with the process analyzing functionality by implementing an XPDL process mining and analysis system, thereby supporting the information control net process discovery functionality. The system seamlessly performs the multifunctionality of discovering, analyzing, and simulating very large scale and massively parallel process models. Almost all of the previous tools and systems for the process analysis and simulation activities could be applied to the discovered and explored process models by the process mining systems. However, it is not easy for the process analyzing activity and the process mining activity to be seamlessly integrated with each other.

- Firstly, we try to investigate the state-of-the-art in the process analysis and simulation tools and systems, separately. The authors of [6] defined the concept of the structured process model and its properties, which are the properties that can be checked up by the system proposed in this paper, and described a taxonomy for analyzing unstructured processes, which are characterized by the properties of improper nestings or mismatched split-join pairings.
- In [4,8], the authors proposed a functional mechanism for analyzing XPDL process models and their relevant data modification sequences, and we use the functional mechanism for implementing the simulation-based performative analysis function of the system in this paper.
- The authors of [18] gave a definition of the system of this paper, in which the authors described a template that was built in the simulation language, Arena. The positive effect of using the template is decreasing the gap between the conceptualization activities and their real process models to be analyzed.
- The necessity of the system is given by [19], in which the authors emphasized the importance of the comparison analysis between designed processes and redesigned processes by using the process analysis and simulation tools and systems. The authors also discussed a number of analysis tools that are relevant for the business process field, evaluated their applicability for business process analysis and simulation, and formulated the technical recommendations and further research issues. The system proposed in this paper is based upon the theory of the information control net modeling methodology.
- Alternatively, Ref. [7] presented a tool of process analysis and simulation based upon the theory of Petri net modeling methodology, which is named Yasper [7] as a tool for modeling, analyzing, and simulating process models based on Petri nets.
- The authors of [20] proposed a fully automatic method to simplify BPML (business process model and notation) process models and described a two-phase iterative algorithm to achieve this simplification, which follows a heuristic approach that makes intensive use of a pattern repository.
- The study of [21] introduced a goal-driven process evaluation method based on process mining for healthcare processes. The proposed method comprises the following steps: defining goals and questions, data extraction, data preprocessing, log and pattern inspection, process mining analysis and generating answers to questions, evaluating results, and initiating proposals for process improvements. Additionally, the authors applied the proposed method to the surgical process of a university hospital in Turkey as a case study.
- The authors of [22] proposed three strategies (based on exhaustive search, genetic algorithms, and a greedy heuristic) that use event data to automatically derive a process model from a configurable process model that better represents the characteristics of the process in a specific

branch. Additionally, they implemented these strategies and tested them in both business-like event logs, as recorded in a higher educational enterprise resource planning system, and a real case scenario involving a set of Dutch municipalities.

- In [23], the authors introduced the method of Accimap from the discipline of accident analysis to analyze the diagnosis results of process mining by creating a complaint handling service process management Accimap model and using it across different system levels. Additionally, they performed a case study in a big manufacturing company in China to verify the proposed method and approach. The case study identified 42 complaint handling process management factors and created the complaint handling process management Accimap model as a final outcome.

- In the study of system inference, the authors of [2] proposed an integrated approach with process mining and fuzzy methods to build a system structure of the fuzzy discrete event system specification (Fuzzy-DEVS) model from system behavior. The proposed approach consists of three stages: (1) extraction of event logs from data by using the system entity structure method; (2) discovery of a transition system, using process discovery techniques; (3) integration of fuzzy methods to automatically generate a Fuzzy-DEVS model from the transition system. Finally, it took a plugin in the process mining framework (ProM) environment for inferring a Fuzzy-DEVS model from an event log dataset and carried out a simulation by using the SimStudio tool.

- Other research outcomes [24,25] based on the Petrinet-based process modeling and analyzing methodology, such as those from the research group of the Eindhoven first and now of the Aachen University, from the research group of the Melbourne School of Engineering, and from the research group of the University of Camerino, provided us the essential intuitions and the functional scopes and definitions for specifying the very useful functional requirements and specifications of the system implemented in this paper.

Conclusively speaking, it might be the first trial in the literature so far for seamlessly integrating the SICN process analysis functionality with the SICN process mining functionality. If those SICN process models, which are characterized by the properties of very large scales and massively parallel structures, are discovered from process enactment event log datasets, then this seamless integration is much more important and meaningful in terms of performing the verification analysis and checking simulation-based performative properness. We assume that the original process models embedded in the corresponding process enactment event log datasets are unknown.

## 3. An Integrated Functional Architecture

So far, we suggested the necessity of the conceptual integration with the analysis function of the Buidtime subsystem and the discovery function of the Miningtime subsystem. Based upon that clear and reasonable conceptual goal, we need to devise a concrete integration approach and architecture in this section. Additionally, we describe the details of implementing the devised integrated architecture as a system. Consequently, in the next consecutive subsections we excogitate the details of the concrete goal and functional scope for developing the integrated architecture and system supporting the process mining functionality and the sophisticated process analyzing functionality.

Figure 1 illustrates a situational snapshot of the process-aware enterprise and organization working with the Miningtime subsystem supporting the process discovery and rediscovery functions [26,27], and the Buildtime subsystem supporting the structural analysis functions and the visual-simulation functions. Particularly, we emphasize that the discovery is to explore business-activity processes from the event logs of the executions of the traditional information systems, while the rediscovery is to do mining of the enacted business-activity processes from the event logs and traces stored whenever the corresponding business-activity processes are enacted and executed by their process management system. In this section, we focus on proposing a concrete integration approach of

the rediscovery function through the Miningtime subsystem and the analysis and simulation functions through the Buildtime subsystem.

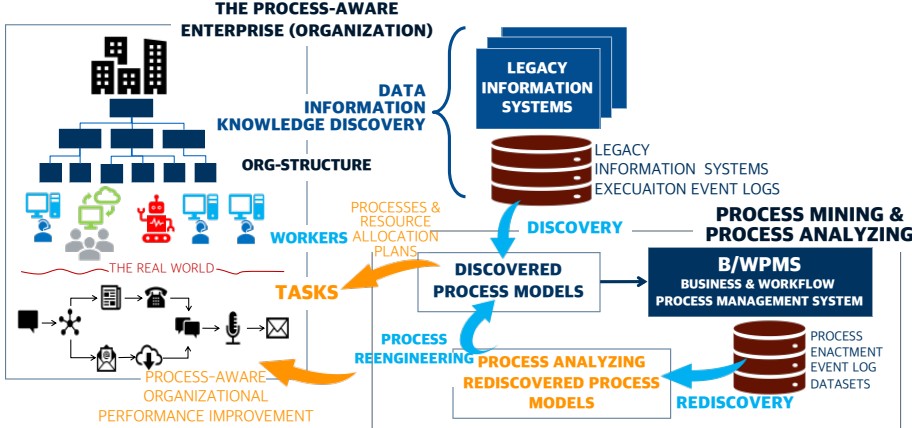

**Figure 1.** Mining and analyzing activities on a process-aware enterprise: the rationale of functional integration of process mining and process analyzing.

Basically, the functional integration of the process mining and the process analyzing is to pursue seamless operations from the rediscovery functions to the simulation-based analysis functions. More precisely, the process mining is to rediscover process models from process enactment event log datasets. The process analyzing is to fulfill the structural and simulation-based analysis on those rediscovered SICN process models. Note that in this paper we especially focus on the process visual-simulation analysis that supports the simulation-based performative properness on the rediscovered process models. The rediscovered process models of the process mining are represented in a mathematical form of the structured information control nets (SICN) [14] and in a graphical form of the GraphML [28] file format with the Graphviz graph visualization open source library. Our experimental cases are also based upon those SICN process models discovered from XES-formatted datasets of process enactment event logs and traces, which are chosen from the 4TU.Centre for Research Data [17]. Therefore, the functional integration between the process mining and the process analyzing ought to be realized seamlessly through transforming the graphical form of Graphviz and GraphML into the textual form of XPDL [10] of the discovered SICN process models.

The overall integrated architecture and its components are illustrated in Figure 2. The upper part of the figure shows an integrated architecture consisting of the process mining subsystem and the process analyzing subsystem, and the lower part shows the functional components of the integrated architecture and system, such as report generators, structural analyzers, a performative simulator, and a visualizer. In particular, the XPDL process instance tracing functionality of the performative simulator ought to be a focal function that is provided by the integration of the process mining and the process analyzing. Conclusively, the scope and goal of the paper is to develop the XPDL process's structural analyzer and performative simulator in conjunction with the SICN process mining system. In the consecutive sections, we describe the design and implementation details of these architectural components. The eventual goal of the integrated architecture is to analyze the structural and behavioral aspects of a SICN process model rediscovered from an enactment event log dataset.

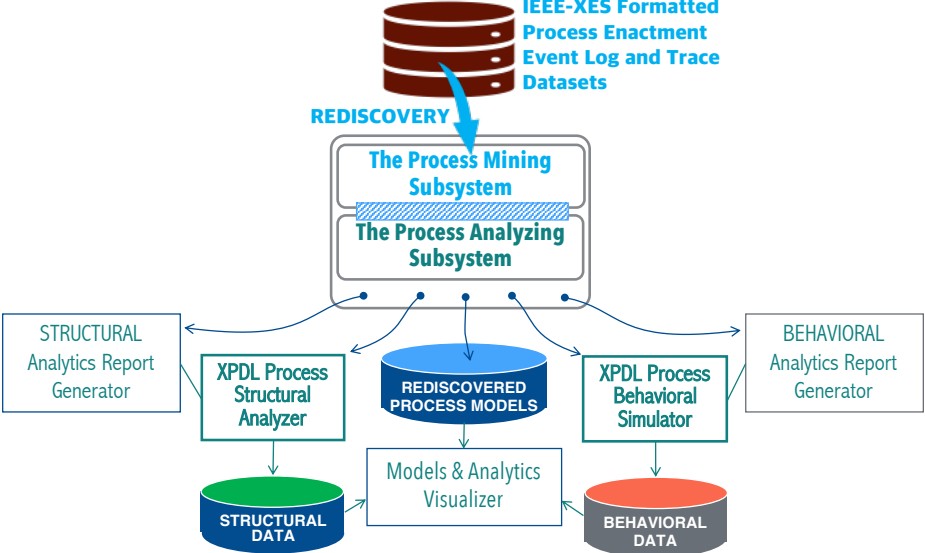

**Figure 2.** The ultimate architecture of the functional integration of process mining and process analyzing.

## 4. Process Mining Functionality

The crucial advantage of using the analysis functionality being conjunct with the mining functionality ought to be that we can automatically analyze such rediscovered process models. Especially, when the underlying process models of the datasets are unknown and these rediscovered process models are very large scale and massively parallel process models, the users can realize the real values and merits of the process analysis functionality. This subsection describes the architectural details of the process mining functionality as a process rediscovery subsystem (Miningtime) that provides the input to the process analysis subsystem (Buildtime).

### 4.1. Structural and Behavioral Process Patterns

The structured information control net (SICN) process model [6,14] is a mathematical and graphical graph model to build the theoretical formalism of process-aware business activities. It is well known as the fittest methodology to describe and analyze control and information flows among the process-aware business activities by capturing temporal transitions and associations among the essential entities, such as business activities, roles, actors, applications, and repositories. Furthermore, it has been used within actual and hypothetical automated offices to yield a comprehensive description of activities, to test the underlying office description for certain flaws and inconsistencies, to quantify certain aspects of office information flows, and to suggest possible office restructuring permutations. In this subsection especially, we focus on the process-centered (control-flow) structural attributes in a process model of the structured information control nets.

The structural and behavioral complexity of a SICN process model is determined by the control-flow attributes that are built through the combinational permutation of the four types of the structural transition patterns, such as sequential, disjunctive, conjunctive, and iterative transition patterns, as graphically shown in Figure 3 and formally defined in Definition 1. That is, the structure of a SICN process model is built by a set of business activities connected by temporal orderings (control flow) called activity transitions. The business activities can be temporally related to each other by combining sequential transition type, disjunctive transition type (after activity $\alpha_A$, do activity $\alpha_B$, or $\alpha_C$ alternatively) with predicates attached, conjunctive transition type (after activity $\alpha_A$, do activities $\alpha_B$ and $\alpha_C$ concurrently), and iterative transition type. Naturally, the composite transition types (except the sequential transition type) are represented by the so-called gateway activity types with split and

join properties. In the SICN process model, the gateway activity types must keep the properties of matched pairings and proper-nestings with splits and joins.

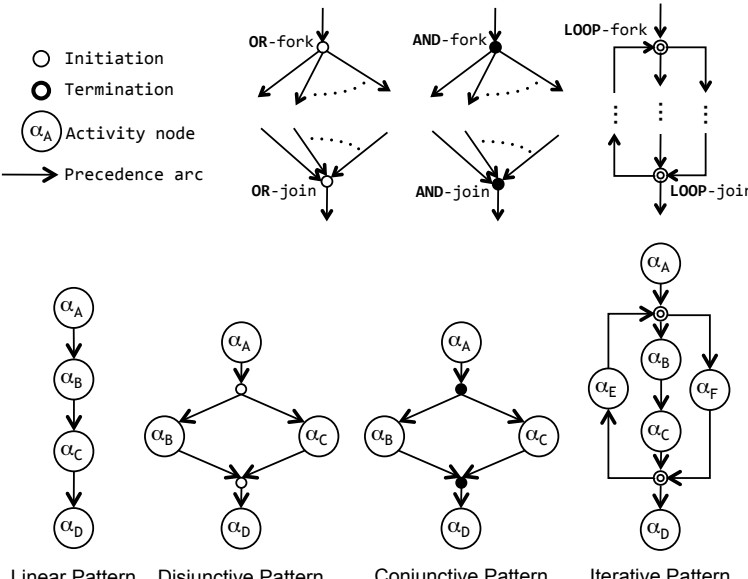

**Figure 3.** The structural control-flow primitives in structured information control net process models.

**Definition 1.** *Structural control-flow patterns in structured information control nets. The basic control-flow structure of a SICN process model is formally defined through 4-tuple* $\Gamma = (\delta, \kappa, I, O)$ *over a set of A activities (including a set of group activities), a set T of transition conditions, where*

— **I** *is a finite set of initial input repositories, assumed to be loaded with information by some external process before execution of the model;*

— **O** *is a finite set of final output repositories, which contains information used by some external process after execution of the model;*

— $\delta = \delta_i \cup \delta_o$*: control-flow structural attributes*
 *where,* $\delta_o : A \longrightarrow \wp(A)$ *is a multivalued mapping function of an activity to its set of (immediate) successors, and* $\delta_i : A \longrightarrow \wp(A)$ *is a multivalued mapping function of an activity to its set of (immediate) predecessors;*

— $\kappa = \kappa_i \cup \kappa_o$*: transition-condition associative attributes*
 *where* $\kappa_i : T \longrightarrow \wp(A)$ *is a multivalued mapping function of an activity* $(\alpha \in A)$ *to its incoming transition-conditions on each arc,* $(\delta_i(\alpha), \alpha)$*; and* $\kappa_o : T \longrightarrow \wp(A)$*: is a multivalued mapping function of an activity* $(\alpha \in A)$ *to its outgoing transition-conditions on each arc,* $(\alpha, \delta_o(\alpha))$*.*

*4.2. The Process Mining Approach: ρ-Algorithm*

This subsection simply describes a process mining framework that is proposed and implemented by the authors' research group. As shown in Figure 4, the framework is able to rediscover all the four types of primitive process patterns in a structured information control net process model from a process enactment event log dataset by applying the concept of mass with their enactment event log occurrences. The conceptual initiation of the process mining framework is introduced in this subsection. Especially, the core component of the framework is the ρ-Algorithm [26,29]. In the ρ-Algorithm, the symbol and name of *rho* (ρ) comes from the A programming language (APL) firstly released in 1960s. The function ρ, coded like ρX in APL, implies that it gives the number of elements in X, from which the concept of **mass** comes. The central idea of the rediscovery algorithm of the framework is exactly the same for the implication of the APL function, rho (ρ). The ρ-operator gives the masses (occurrences) of all the activities in the event log dataset of a corresponding process

model. In the paper, the emphasis is placed on rediscovering very large scale and massively parallel process models that are usually built by a combination of the four types of primitive process patterns, linear, disjunctive (exclusive-OR), conjunctive (parallel-AND), and repetitive (iterative-LOOP) process patterns, which were formally defined in the previous section.

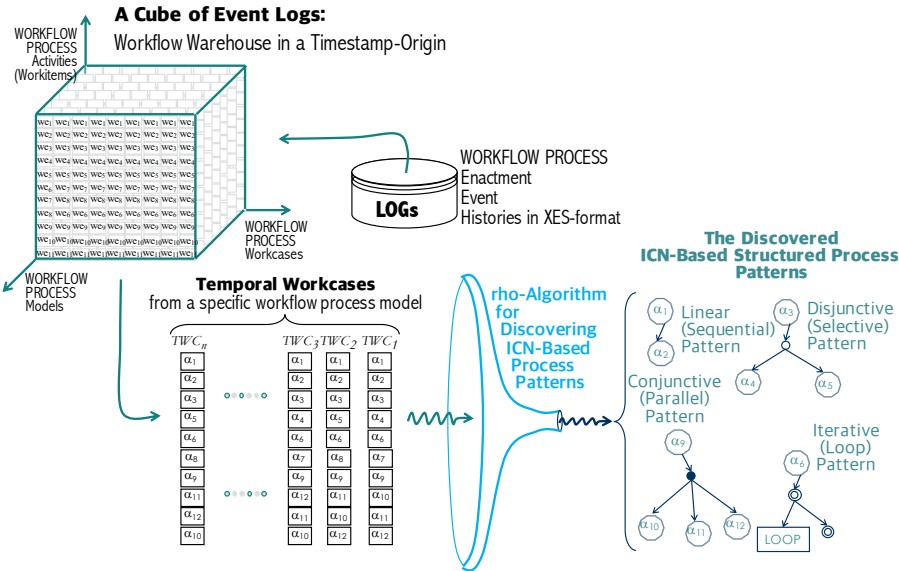

**Figure 4.** The process mining framework.

The overall algorithmic approach is a stepwise mining procedure with a series of graphical transformations to be used for rediscovering all the types of the primitive process patterns that constitute a structured information control net process model. That is, as shown in Figure 5, the $\rho$-Algorithm consists of three major transformation steps, STEP-1, STEP-2, and STEP-3, from forming temporal workcases out of the process enactment event logs to discovering a structured information control net process model as a result. The first transformation of the algorithm is to discover the enacted workcases from the event logs, each of which can be modeled into a temporal workcase model. At the same time, it is necessary to count the occurrence of each temporal workcase with its activities. The second transformation of the algorithm is to discover an activity-driven pattern graph by integrating all the members of the adjacent-activity set and calculating the occurrences of the temporal workcases. In terms of rediscovering the structured information control net process model from the corresponding activity-driven pattern graph, we develop the third transformation of the algorithm that is able to transform to any combinational number of AND/OR/LOOP primitive process patterns.

- *STEP-1: Groups of Temporally Ordered Adjacent-Activity Pairs*: The first step of the $\rho$-Algorithm is to mine a group of temporally ordered adjacent-activities pairs from temporal workcases of the process instance event logs. Additionally, each of the temporal workcases is formally represented by one of the workcase model types introduced in the conceptual framework. That is, a temporal workcase represents an ordered enactment sequence of activity event logs, each of which is formed with its activity identifier and its time-stamp extracted from its corresponding process enactment event log.

- *STEP-2: Quantitative Adjacent-Activity Set and Process Pattern Graph*: The STEP-2 of the $\rho$-Algorithm is to build all the groups of temporally ordered adjacent-activity pairs, each of which corresponds to a process instance event trace. The eventual output of this algorithm is a quantitative adjacent-activity set named adjacencyList $\beta$. This set is built from all the groups of temporally ordered adjacent-activity pairs through an internal transformation procedure.

- *STEP-3: Rediscovering Structured Process Patterns*: The final step (STEP-3) of the $\rho$-Algorithm is to discover all the primitive process patterns of a structured information control net process model from the mined process pattern graph discovered from all the groups of temporally ordered adjacent-activity pairs. The eventual goal of the $\rho$-Algorithm is accomplished through this step. Note that the structured information control net process model must be satisfied with the proper nesting and the matched pairing properties in forming gateway activities in each primitive process graph pattern.

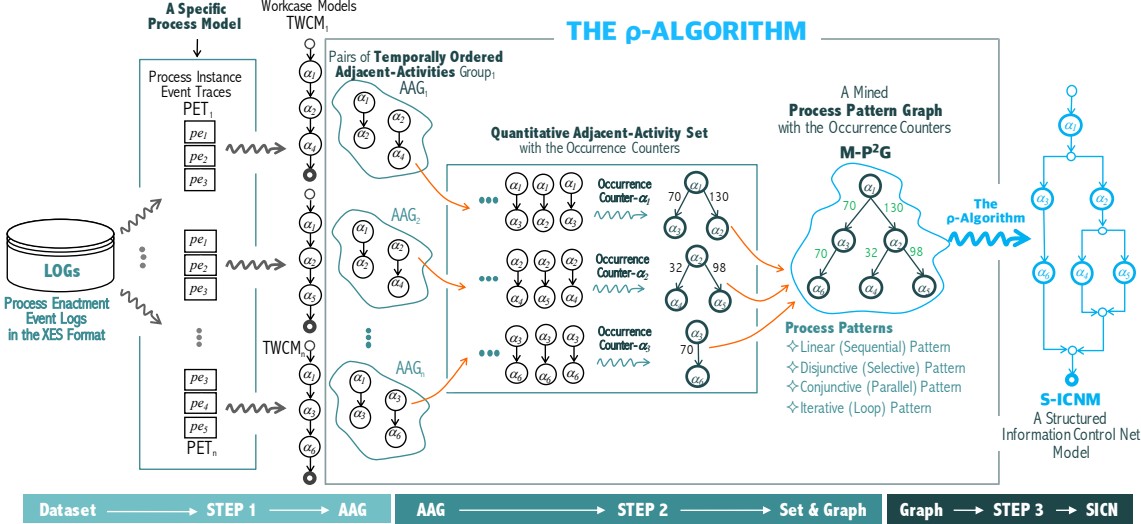

**Figure 5.** The $\rho$-Algorithm with three steps of graphical transformations.

Based upon the $\rho$-Algorithm, the authors' research group has successfully developed a process mining system that is able to rediscover a structured information control net process model coping with all the four types of process patterns from a dataset of process enactment event logs. By using the process mining system, this paper carries out a series of experimental analyses by deploying the implemented system onto a dataset provided by the BPI challenges of 4TU.Centre for Research Data [17] and formatted in the IEEE XES standardized format [30]. Summarily, the following are the characteristics of the $\rho$-Algorithm and the process mining framework:

- First, the process mining framework is able not only to rediscover the process patterns but also to discover the enactment proportions [27] of the process patterns from a dataset of the IEEE XES-formatted enactment event logs of a corresponding process model.
- Second, the process mining framework is theoretically supported by the information control nets modeling methodology [16] of process models.
- Third, the essential algorithm of the process mining approach is named the $\rho$-Algorithm (rho-Algorithm) that is able to rediscover a structured information control net model with the enactment occurrences of the activities associated with an underlying process model.
- Fourth, the $\rho$-Algorithm was firstly developed in the process management and mining literature as a process mining algorithm that discovered a structured process model theoretically supported by the structured information control net modeling methodology.
- Fifth, the $\rho$-Algorithm is able to rediscover all the primitive process patterns, such as linear (sequential), conjunctive (parallel-AND), disjunctive (exclusive-OR), and repetitive (iterative-LOOP) process patterns, and discover the enactment occurrences and proportions of each branch of the process patterns.

## 5. Process Analyzing Functionality

Once a SICN process model is rediscovered from a dataset, then the model is analyzed by the process analyzer, which is architecturally composed of a structural analyzer and a visual simulator, to be implemented in the paper after being transformed into its XPDL process model. Therefore, this subsection describes the details of the algorithms of the structural analyzer and the visual simulator. The structural analyzer is based on the standardized formats of XPDL and extracts the structural attributes of the rediscovered SICN process model, which are defined in the first subsection. The next consecutive subsections describe the architectural and functional components of the process analyzer, which are mainly focused on the structural aspect and the behavioral aspect analyzed by the structural analyzer and the visual simulator, respectively.

### 5.1. Architectural Components of the Process Analyzer

This subsection describes the detailed design specifications of the process analyzer with supporting structural and behavioral analysis in a fashion of visual simulation. As already illustrated in Figure 2, the functional architecture of the process analyzer is composed of two groups of architectural components. One is the XPDL process structural analyzer and its related components; the other is the XPDL process performative simulator and its related components. Actually, there are five functional components with three database connection agents: the model and analytics visualizer, the XPDL process structural analyzer, the XPDL process performative simulator, the structural analytics report generator, and the performative analytics report generator. The database schema for the process analyzer consist of rediscovered XPDL process models' database schema, structural database schema, and performative database schema. As we know, the rediscovered process model database schema are for storing XPDL process models rediscovered and transformed by the process mining functionality described in the previous subsection. The structural database schema and the performative database schema preserve the structural analysis and the behavioral analysis results, respectively, including the statistical data of the structural attributes and the behavioral attributes of those XPDL process models.

The *rediscovered process models and analytics visualizer* are in charge of the overall control of the analyzer, including user management, session and access control, and XPDL process management, in particular. Assume that the rediscovered process model is based upon the structured information control net methodology, and the process mining subsystem supports the *Export-to-XPDL* function that automatically translates the graphical representation of a rediscovered SICN process model into a textual representation of its corresponding XPDL process model. Since then, all the functions and operations, such as verifying, analyzing, reporting, and visualizing operations, to be applied into the XPDL process models, are controlled and managed via this visualizer. The *structural and performative report generators* are to generate and visualize the structural analysis and performative simulation results and statistics of each structural component in an XPDL process model. This generator is able to produce two-level structural and performative simulation results and statistics. The first level is the process-level analytical results and statistics, and the other is the package-level analytical results and statistics. For the sake of providing the complete version of process analyzing functionality, the *XPDL process structural analyzer* and *XPDL process animator* including the *XPDL process instance tracer* perform the following verifiable analysis functions:

- Structural properness verification: checking up whether the rediscovered process model is keeping the rules of proper-nestings and matched pairings when building its gateway-type activities.
- Associative relationship verification: checking up whether the rediscovered process model is keeping the correct associative attributes when building activity-to-role associations, activity-to-program associations, activity-to-data associations, and role-to-actor associations.

- Performative properness verification: checking up whether the rediscovered process model is maintaining efficiency when managing the dynamic aspects at the package level, process level, activity level, and the performer level (performative indicators).
- Behavioral simulation verification: based upon the rediscovered process model with its enactment event histories, checking up the enactment histories by tracing a process instance of the corresponding model via step-by-step clicking operations.

### 5.2. Generation of the Rediscovered XPDL Process Models

The initiation step of the process analyzing approach is to perform a transformation function that generates the rediscovered XPDL process model from the graphical form of the rediscovered SICN process model. The algorithm of this transformation step is represented in Algorithm 1 named the XPDL process model generation algorithm. Note that the tagged constructs of XPDL are described in the next section of the process analyzing functionality. As you can see in the Algorithm 1, it requires an activity graph of the rediscovered SICN process model, which is the internal graph model of the $\rho$-Algorithm, as input ($\Gamma$), and ensures the XPDL process model corresponding to the rediscovered SICN process model, as output ($\gamma$), finally. The XPDL process model plays very important role as a bridge between the process mining functionality and the process analyzing functionality. Therefore, the focal point of the functional integration of the process mining and the process analyzing ought to be this algorithm and its output, the XPDL process model, because it is possibly realized to complete the seamless integration through this algorithm and its output.

### 5.3. The Structural Analysis of Rediscovered XPDL Process Models

This subsection describes the structural attributes used in rediscovered XPDL process models, the specifications of which were released by the International Standardization Organization of Workflow Management Coalition. The XPDL version 1.0 is the XML-formatted version of the workflow process definition language (WPDL). Recently, the workflow management coalition released the new specifications of XPDL reflecting the OMG's standardized graphical notation of the process modeling notation (BPMN) as the XPDL version 2.0. In this paper, we consider the XPDL version 2.1 as the textual format of a SICN process model, and it will be the input of the structural analyzer. In particular, the control-flow structural attributes can be formed by the from and to properties in the transition attribute. Conclusively, the structural attributes of the XPDL version 2.1 are summarized in Table 1. Note that, because of the page limitation, we simply introduce the structural attributes and their properties in this paper.

Based upon the structural attributes and their occurrences in a rediscovered XPDL process model (or package), the XPDL process structural analyzer is able to produce the two-level analytical statistics, such as the package-level analytical statistics and the process-level analytical statistics. At the package-level, it supports the activity-type structural analysis, component-type structural analysis, component-type's usage ratio analysis, and the subprocess usage ratio analysis. It also supports the activity-related structural attributes analysis, and the associative relationship attributes analyses, such as activity-to-program, activity-to-role, activity-to-data, and role-to-performer associations. Note that a process package comprises a group of process models, and the XPDL schema is formatted from a pair of the package-level tags, such as `<package> ... </package>`. The following are the analytical statistics to be analyzed and produced by the structural analyzer, which ought to be extensively represented in the rediscovered XPDL process models through their corresponding extended tags:

(1) Package-level Structural Analysis Statistics

- The number of process models in a corresponding process package;
- The number of activities in each model and their usage ratios;
- The number of roles in each model and their involvement ratios;

— The number of invoked applications in each model and their usage ratios;

— The number of relevant data types in each model and their usage ratios;

— The number of subprocesses in each model and their usage ratios;

— The usage ratio of each model as subprocesses.

(2) Process-level Structural Analysis Statistics

— Species of structural patterns and their usage ratios;

— The number of participants (actors or performers) and their participation ratios;

— The number of roles and their involvement ratios;

— The number of invoked applications and their usage ratios;

— The number of relevant data types and their usage ratios;

— The number of subprocesses and their usage ratios.

---

**Algorithm 1** The XPDL process model generation algorithm.

---

**Require:** An activity graph of a rediscovered SICN process model, $\Gamma$;

**Ensure:** The XPDL process model, $\gamma$ of $\Gamma$;

 1: Initialize $\gamma \leftarrow \varnothing$;

 2: $\gamma.InsertHeader()$; ▷ Insert the header format of XPDL ver 2.1: <xpdl:Pakage...>, <xpdl:PackageHeader>

 3: $\gamma$.InsertTagAttribute("xpdl:Processes");         ▷ Insert the processes attribute to $\gamma$

 4: $\gamma$.InsertTagAttribute("xpdl:Activities");         ▷ Insert the activities attribute to $\gamma$

 5: **for** ( $\forall$ node $\eta$ in $\Gamma$ ) **do** ▷ Read all the nodes (actvities or gateways) in $\Gamma$ and then add all of their tags to $\gamma$

 6:   $\gamma$.InsertTagAttribute("xpdl:Activity");       ▷ Insert the activity attribute to $\gamma$

 7:   currentNode $\leftarrow \eta$.GetCurrrentNodeInfo();

 8:   routeType $\leftarrow$ defaultRouteType; shapeType $\leftarrow$ defaultShapeType; gateType $\leftarrow \varnothing$;

 9:   **if** (currentNode is OrGateway) **then**

10:    routeType $\leftarrow$ "Exclusive-OR";

11:    shapeType $\leftarrow$ "Exclusive-OR Gateway";

12:   **else if** (currentNode is AndGateway) **then**

13:    routeType $\leftarrow$ "Parallel-AND";

14:    shapeType $\leftarrow$ "Parallel-AND Gateway";

15:   **else if** (currentNode is LoopGateway) **then**

16:    routeType $\leftarrow$ "Iterative-LOOP";

17:    shapeType $\leftarrow$ "Iterative-LOOP Gateway";

18:   **end if**

19:   **if** (currentNode is OpenGateway) **then**

20:    gateType $\leftarrow$ "Split";

21:   **else if** (currentNode is ClosedGateway) **then**

22:    routeType $\leftarrow$ "Join";

23:   **end if**

24:   $\gamma$.InsertAttribute(type=activity, currentNode, routeType, shapeType, gateType);

25: **end for**

26: $\gamma$.InsertTagAttribute("xpdl:Transitions");       ▷ Insert the transition attribute to $\gamma$

27: **for** ( $\forall$ *edge* $\theta$ in $\Gamma$ ) **do**

28:   currentEdge $\leftarrow \theta$.GetEdgeInfo();

29:   fromActivityID $\leftarrow$ currentEdge.GetFromID();

30:   toActivityID $\leftarrow$ currentEdge.GetToID();

31:   $\gamma$.InsertAttribute(type=transition, fromActivityID, toActivityID);

32: **end for**

33: **Return** $\gamma$;          ▷ Finally output the XPDL process model

---

**Table 1.** Structural attributes in the XML process definition language.

| Package | Process | Activity | Transition | Application | Data-Field | Participant |
|---|---|---|---|---|---|---|
| -Identifier<br>-Name<br>-Descrip.<br>-Ex_Attr. | -Identifier<br>-Name<br>-Descrip.<br>-Ex_Attr. | -Identifier<br>-Name<br>-Descrip.<br>-Ex_Attr. | -Identifier<br>-Name<br>-Descrip.<br>-Ex_Attr. | -Identifier<br>-Name<br>-Descrip.<br>-Ex_Attr. | -Identifier<br>-Name<br>-Descrip.<br>-Ex_Attr. | -Identifier<br>-Name<br>-Descrip.<br>-Ex_Attr. |
| -XPDL<br>-Source<br>-Creation<br>-Version<br>-Author<br>-Country<br>-Publ_St<br>-Conform<br>-Priority | -Cre_Date<br>-Version<br>-Author<br>-Code_P<br>-Country<br>-Publ_St<br>-Priority<br>-Limit.<br>-Val_Fr<br>-Val_To | -A_Mode<br>-Split<br>-Join<br>-Priority<br>-Limit.<br>-St_Mode<br>-Fi_Mode<br>-Deadline | | | -Data_Ty | -Par_Ty |
| -Respon.<br>-Ext_Pac | -Params.<br>-Respon. | -Perf.<br>-Tool<br>-Subflow<br>-Act_Set<br>-A_param | -Conditi.<br>-From_Tr<br>-To_Tr | -Params. | -Ini_Val. | |
| -Docum.<br>-Icon | -Docum.<br>-Icon | -Docum.<br>-Icon | | | | |
| -Cost_Unt | -Dur_Unt<br>-Duration<br>-Wait_T<br>-Work_T | -Cost<br>-Duration<br>-Wait_T<br>-Work_T | | | | |

*5.4. The Behavioral Analysis of Rediscovered Process Models*

The behavioral analysis of a rediscovered XPDL process model is based upon the primitive control-flow process patterns introduced in [Definition 1] in the previous subsection. The control-flow attributes of the rediscovered process model are also composed by the combinational permutation of the four types of the structural transition patterns, such as sequential, disjunctive, conjunctive, and iterative transition patterns. Therefore, the behaviors (i.e., execution sequences of process instances) of the rediscovered process model are analyzed and measured by tracing these combinational primitive process patterns. Figure 6 illustrates how the behavioral analysis and statistical measurement works on the rediscovered process model. The $\rho$-Algorithm rediscovers a SICN process model from a dataset of event logs. Assume that the rediscovered SICN process model is made up of two disjunctive process patterns and a single conjunctive process pattern, as shown in the figure. The algorithm discovers not only the SICN process model but also the execution occurrences of its process instances and workitems, in addition. From the rediscovered SICN process model and its execution statistics, the process analyzing function generates the corresponding XPDL process model with extended tags of the statistical attributes and analyzes the behavioral sequences and quantitative statistical measurement, as well. [Definition 2] defines the behavioral sequence net; that is, the formal representation of a behavioral sequence and its execution statistics (i.e., the number of occurrences). Additionally, the Algorithm 2 named the behavioral sequence analysis algorithm is for generating a set of behavioral sequence nets from a rediscovered SICN process model. Note that the algorithm deals with only three process patterns, including sequential, conjunctive-AND, and disjunctive-OR process patterns, thereby excepting the iterative-LOOP process pattern, because the iterative-LOOP process pattern is treated as a special type of disjunctive-OR process pattern that ought to be properly dealt with by the disjunctive-OR routines of the algorithm. Practically, the XPDL standards do not specify the iterative-LOOP transition type as a normal process pattern either.

---

**Algorithm 2** The behavioral sequence analysis algorithm.

---

**Require:** A rediscovered SICN process model, $\Gamma$;

**Ensure:** A Set of Behavioral Sequence Nets, $\Omega$ of $\Gamma$;

1: initialize $\Omega \leftarrow \varnothing$;

2: **Procedure BSAFunc ( in** $s \leftarrow \{\alpha_I\}, \Omega$ **)**;          $\triangleright$ Recursive procedure for discovering $\Omega$

3: **begin**

4:   $v \leftarrow s; \Omega.A^{bs} \leftarrow \Omega.A^{bs} \cup \{v\}$;

5: **while** ( $u \leftarrow \delta_o(s)\ != \alpha_F$ ) **do**

6:     **if** ( $u$ is a sequential activity? ) **then**

7:        $w \leftarrow u; \Omega.A^{bs} \leftarrow \Omega.A^{bs} \cup \{w\}$;

8:        $\Omega.\varrho_o(v) \leftarrow w; \Omega.\varrho_i(w) \leftarrow v; \Omega.\beta_o(v) \leftarrow \kappa_o(s); \Omega.\beta_i(v) \leftarrow \kappa_i(s)$;

9:        $\phi_o((v, o \in \varrho_o(v))) \leftarrow$ occurrences of $o \in \varrho_o(v)$;

10:       $\phi_i((\epsilon \in \varrho_i(v), v)) \leftarrow$ occurrences of $v$;

11:     **else if** ( $u$ is a conjunctive AND-split-gateway activity? ) **then**

12:       $w \leftarrow u; \Omega.A^{bs} \leftarrow \Omega.A^{bs} \cup \{w\}$;

13:       $\Omega.\varrho_o(v) \leftarrow w; \Omega.\varrho_i(w) \leftarrow v; \Omega.\beta_o(v) \leftarrow \kappa_o(s); \Omega.\beta_i(v) \leftarrow \kappa_i(s)$;

14:       **for** ( each of $\forall \alpha \in \delta_o(u)$ ) **do**

15:         $x \leftarrow \alpha; \Omega.A^{bs} \leftarrow \Omega.A^{bs} \cup \{x\}$;

16:         $\Omega.\varrho_o(w) \leftarrow w; \Omega.\varrho_i(x) \leftarrow w$;

17:         $\Omega.\beta_o(w) \leftarrow \kappa_o(u); \Omega.\beta_i(w) \leftarrow \kappa_i(u)$;

18:         $\phi_o((w, o \in \varrho_o(w))) \leftarrow$ occurrences of $o \in \varrho_o(w)$;

19:         $\phi_i((\epsilon \in \varrho_i(w), w)) \leftarrow$ occurrences of $w$;

20:       **end for**

21:       **for** ( each of $\forall \alpha \in \delta_o(u)$) **do**

22:         **call Procedure BSAFunc ( in** $s \leftarrow \alpha, \Omega$ **)**;

23:       **end for**

24:     **else if** ( $u$ is a disjunctive OR-split-gateway activity? ) **then**

25:       $w \leftarrow u; \Omega.A^{bs} \leftarrow \Omega.A^{bs} \cup \{w\}$;

26:       $\Omega.\varrho_o(v) \leftarrow w; \Omega.\varrho_i(w) \leftarrow v; \Omega.\beta_o(v) \leftarrow \kappa_o(s); \Omega.\beta_i(v) \leftarrow \kappa_i(s)$;

27:       **for** ( each of $\forall \alpha \in \delta_o(u)$ ) **do**

28:         **call Procedure BSAFunc ( in** $s \leftarrow \alpha, \Omega$ **)**;

29:       **end for**

30:     **else if** ( $u$ is either an OR-join-gateway or an AND-join-gateway activity? ) **then**

31:       $w \leftarrow u; \Omega.A^{bs} \leftarrow \Omega.A^{bs} \cup \{w\}$;

32:       $\Omega.\varrho_o(v) \leftarrow w; \Omega.\varrho_i(w) \leftarrow v; \Omega.\beta_o(v) \leftarrow \kappa_o(s); \Omega.\beta_i(v) \leftarrow \kappa_i(s)$;

33:       $\phi_o((w, o \in \varrho_o(w))) \leftarrow$ occurrences of $o \in \varrho_o(w)$;

34:       $\phi_i((\epsilon \in \varrho_i(w), w)) \leftarrow$ occurrences of $w$;

35:     **end if**

36:     $s \leftarrow u; v \leftarrow w$;

37: **end while**

38: $w \leftarrow u; \Omega.A^{bs} \leftarrow \Omega.A^{bs} \cup \{w\}$;          $\triangleright$ where $u$ is the final activity, $\alpha_F$

39: $\Omega.\varrho_o(v) \leftarrow w; \Omega.\varrho_i(w) \leftarrow v; \Omega.\beta_o(v) \leftarrow \kappa_o(s); \Omega.\beta_i(v) \leftarrow \kappa_i(s)$;

40: $\phi_o((w, o \in \varrho_o(w))) \leftarrow$ occurrences of $o \in \varrho_o(w)$;

41: $\phi_i((\epsilon \in \varrho_i(w), w)) \leftarrow$ occurrences of $w$;

42: **Return** $\Omega$;          $\triangleright$ Finally output a set of behavioral sequence nets

---

**Definition 2.** *Behavioral sequence net of a rediscovered SICN (or XPDL) process model model. Let* $\Omega$ *be a* **BSN**, *a behavioral sequence net that is formally defined as* $\Omega = (\varrho, \kappa, o)$ *over the discovered workitems,* $A^{bs}$, *and the discovered control-transitions,* $T^{bs}$, *where*

- $\varrho = \varrho_i \cup \varrho_o$ *where,* $\varrho_o : A^{bs} \longrightarrow \wp(A^{bs})$ *is a multivalued mapping of a workitem to its immediate successor, and* $\varrho_i : A^{bs} \longrightarrow \wp(A^{bs})$ *is a multivalued mapping of a workitem to its immediate predecessor;*
- $\beta = \beta_i \cup \beta_o$ *where,* $\beta_i : A^{bs} \longrightarrow T^{bs}$ *is a single-valued mapping of a workitem to its connected control-transition,* $\tau$, $(v \in \varrho_i(\alpha), \alpha) \in T^{bs}$; *and* $\beta_o : A^{bs} \longrightarrow T^{bs}$ *is a single-valued mapping of a workitem to its connected control-transition,* $\tau$, $(\alpha, v \in \varrho_o(\alpha)) \in T^{bs}$, *where* $\alpha \in A^{bs}$;
- $o = o_i \cup o_o$ *where,* $o_i : T^{bs} \longrightarrow N$ *is a single-valued mapping of an incoming control-transition* $(\tau \in \beta_i)$ *to its number of occurrences; and* $o_o : T^{bs} \longrightarrow N$ *is a single-valued mapping of an outgoing control-transition* $(\tau \in \beta_o)$ *to its number of occurrences.*

The behavioral analysis functionality illustrated in Figure 6 and devised in Algorithm 2 is able to produce a set of behavioral sequence nets from the rediscovered SICN/XPDL process model. Our emphasis is placed on the quality analysis of rediscovered process model by using the proposed concept and its algorithm. That is, the behavioral analysis aims at improving the quality of the rediscovered process model. Thus, naturally the following questions are raised: what is the quality of a process model? How can we evaluate the quality of a process model? This paper tries to answer those questions through the concept of behavioral sequence nets supported by the functional integration of the process mining and the process analyzing functionalities. We would insist that the quality of a rediscovered process model be defined by the degree of the discrepancy between the discovered process model as it is built by the Buildtime subsystem and the rediscovered process model as it is rediscovered by the Miningtime subsystem. The discrepancy, as you can easily imagine, is caused by disjunctive and conjunctive process patterns presented on the corresponding process model. The number of disjunctive process patterns on the model will affect the number of *mutually exclusive behavioral sequences*, whereas the number of the conjunctive process patterns on the model will affect the number of *mutually inclusive behavioral sequences*. Additionally, the statistical measurements related to the behavioral sequences play very valuable role in evaluating the quality of the rediscovered process models.

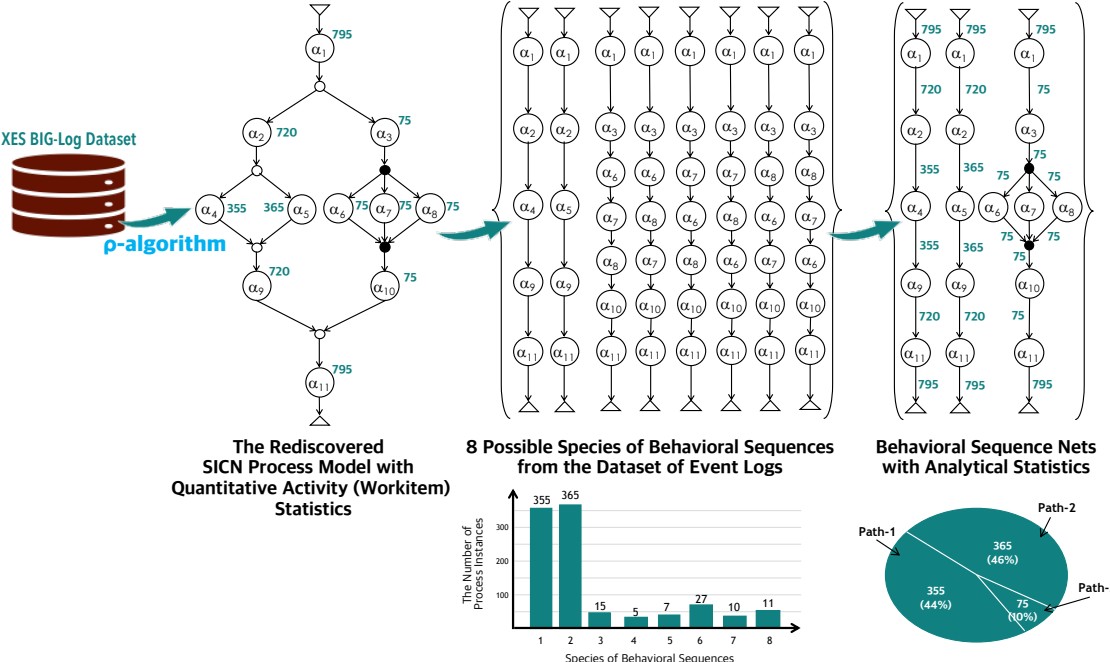

**Figure 6.** The behavioral analysis and statistics of the rediscovered process model.

Each of the behavioral sequences can be traced through the interactive and visual operations provided by the process analyzer. Figure 7 depicts a situational snapshot of performing the interactive and visual analyses through the behavioral sequence tracer. The graph model of the behavioral sequence net was rediscovered from the dataset of the large bank transaction process enactment event logs. Note that the details of the dataset and its rediscovered SICN process model are described in the experimental section of this paper. As shown in the figure, the behavioral sequence net corresponding to the process instance number 0 includes 65 workitems (business activities) with their temporal enactment sequences. Through the interactive and visual analyses on those process instances rediscovered from datasets, we are able to evaluate the properness and quality of the process instances and their models, eventually. The implementation details of the interactive and visual analysis functionality for analyzing the behavioral sequences and its practical applications are described in the experimental studies of the next section.

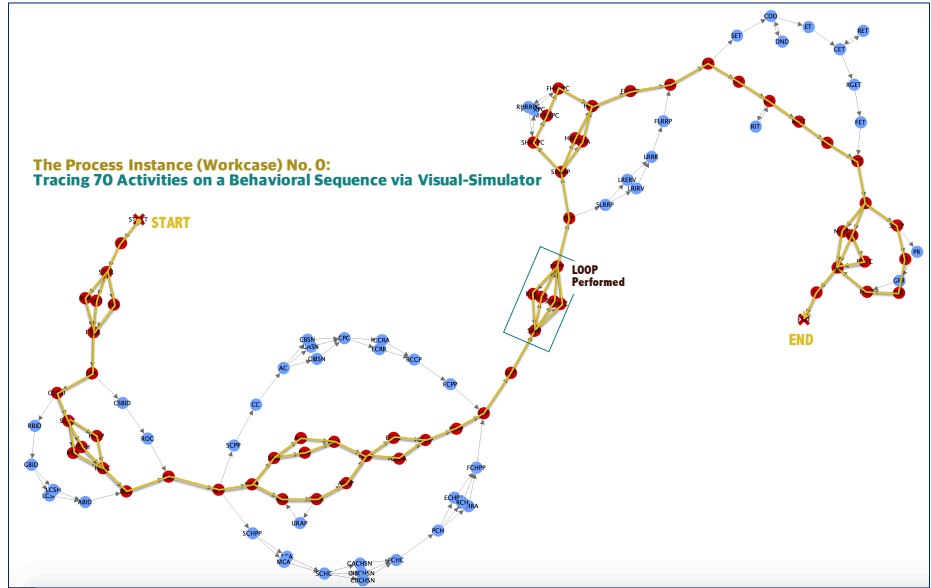

**Figure 7.** The interactive and visual analysis of behavioral sequences of the rediscovered process model.

## 5.5. Implementation of the Integrated Functional Architecture

This section describes the implementation details of the integrated functional architecture designed in the previous section. The major systematic framework is made up of a structural analyzer and a behavioral simulator with a process instance tracer, and these are operable on those rediscovered SICN and XPDL process models from the process enactment event log datasets by the process mining approach developed in this paper. Figure 8 illustrates the concrete systematic framework of the functional integration of the process mining and the process analysis and its experimental results with a specific dataset. As shown in the figure, the process mining functionality was integrated with the process analyzing functionality through seamlessly sharing the XPDL process models transformed from the rediscovered SICN process models. The figure uses a series of captured screens produced through one of the experiments described in the next section. The dataset is in the IEEE XES standardized log format and contains 678,864 event logs recorded from enacting 10,000 workcases (i.e., process instances) of the large bank transaction process model; from the dataset, the process mining subsystem rediscovers the structured information control net process model consisting of 113 activities and their occurrences, and the process analyzing subsystem produces the textual form of the rediscovered XPDL process model; by analyzing the rediscovered XPDL process model, the process analyzing subsystem eventually validates the structural properness and the behavioral properness of the rediscovered process model characterized by the very large scale and massively parallel structural properties.

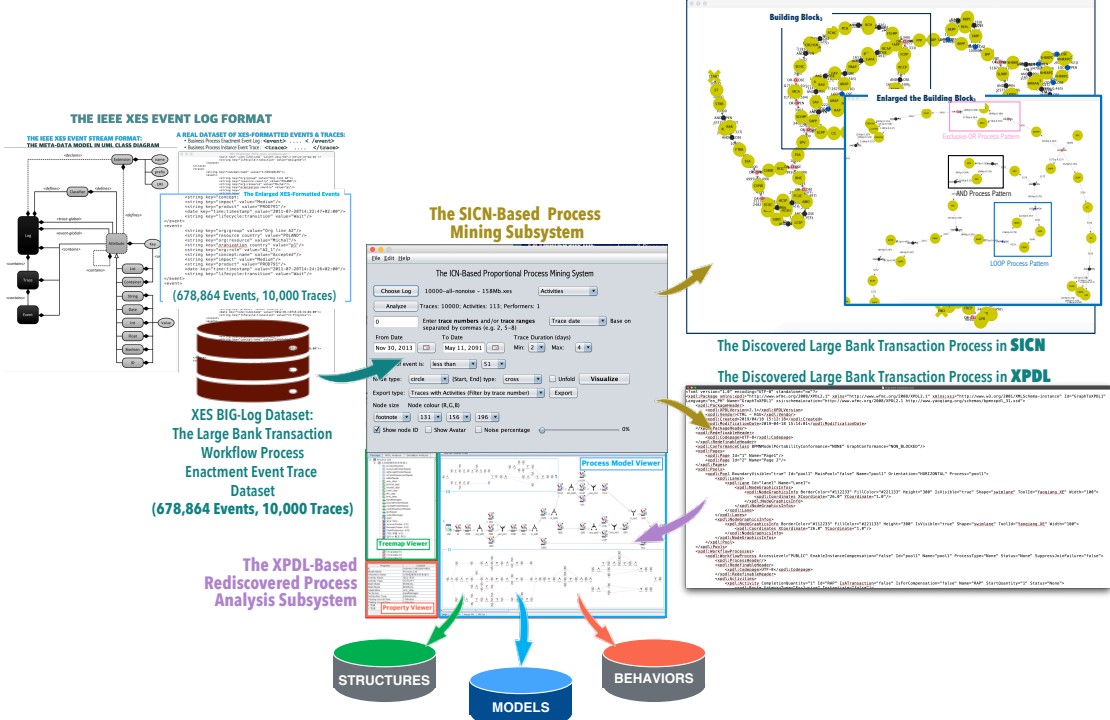

**Figure 8.** The implemented systematic framework of the integrated functional architecture.

## 6. Experimental Validation Studies

This section carries out a couple of feasibility and validation studies of the proposed conceptual architecture: functional integration with the process mining and the process analyzing. One is for the structural analysis; that is, as the implemented conceptual architecture, the systematic framework is implemented and applied to a couple of real datasets obtained from the 4TU.Centre for Research Data [17], each of which is formatted in the IEEE-XES standardized event log format and contains very large scale event logs spawned from enactment histories of very large scale and massively parallel process models. Figure 9 shows the captured screens of the datasets, and the following are the brief descriptions of the datasets used for the experimental validation studies:

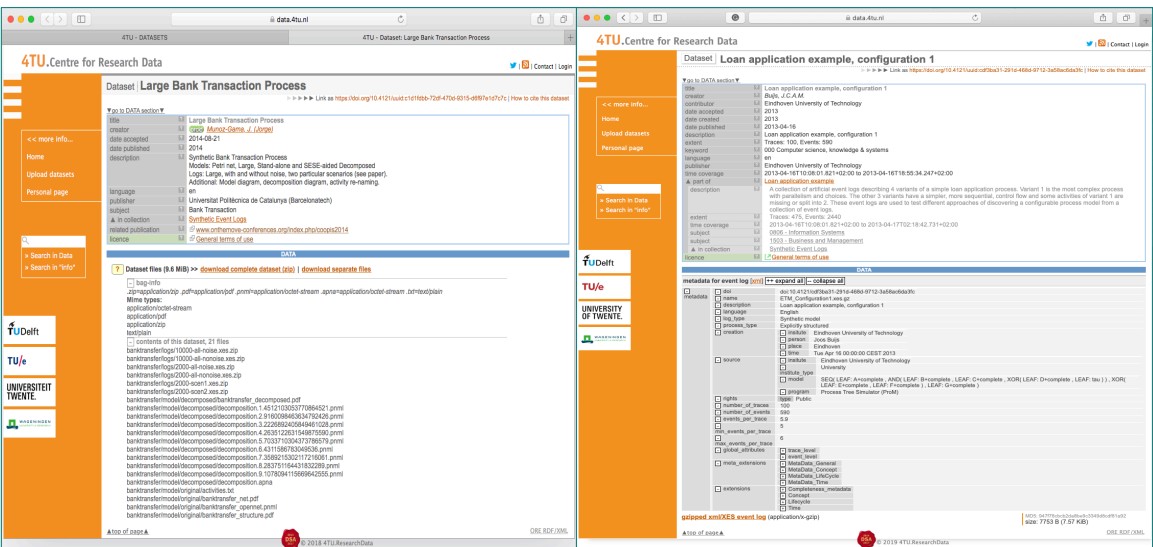

**Figure 9.** The detailed descriptions of the datasets from the 4TU.Centre for Research Data.

— Experimental Dataset-1: Mining and analyzing a structured information control net process model from the enactment event log dataset of the large bank transaction process model:

  – Title: Large Bank Transaction Process;
  – Creator: Munoz-Gama. J. (Jorge);
  – Date accepted and published: 2014-08-21;
  – Description: Synthetic Bank Transaction Process;

    ∗ Models: Petri net, large, stand-alone, and SESE-aided decomposed ;
    ∗ Logs: large, with and without noise, and two particular scenarios;
    ∗ Additional: model diagram, decomposition diagram, and activity re-naming.

— Experimental Dataset-2: Mining and analyzing a structured information control net process model from the enactment event log dataset of the loan application example (Configuration 1) process model:

  – Title: Loan Application Example, Configuration 1;
  – Creator: Buijs, J.X.A.M.;
  – Date accepted and published: 2013-04-16;
  – Description: A collection of artificial event logs describing four variants of a simple loan application process. Variant 1 is the most complex process with parallelism and choices. The other three variants have simpler, more sequential control flows, and some activities of Variant 1 are missing or split into two. These event logs are used to test different approaches of discovering a configurable process model from a collection of event logs.

*6.1. Validation of the Rediscovered SICN Process Models*

First of all, it is necessary to validate the rediscovered process models of structured information control nets (SICN), which are mined from the experimental datasets, respectively. The process mining subsystem of the integrated functional architecture and its implemented system is applied to the Experimental Dataset 1 and the Experimental Dataset 2 and successfully mined the rediscovered SICN process models.

As you can see in Figure 10, the experimental result of the process mining subsystem is the rediscovered SICN process model containing 113 activities with all the primitive process patterns and all the activities' execution occurrences. That is, the left hand side of the figure is a captured screen representing the SICN process model rediscovered from the Experimental Dataset 1 of 10,000 all-nonoise 50MB.xes that contains 10,000 process instance event traces and 678,864 activity event logs under the control of enacting 113 business activities of the large bank transaction process model. The enlarged process block indicates all the primitive process patterns by the labeled boxes, in particular. Moreover, the process mining subsystem supports the printout form of the rediscovered SICN process model, as shown on the right hand side of the figure. Secondly, another experiment of the process mining subsystem was carried out on the Experimental Dataset 2, and it was able to mine the rediscovered SICN process model of the loan application example (Configuration 1), as shown in Figure 11. The captured-screen and its printout form in the figure show the rediscovered SICN process model containing nine activities with their execution occurrences. The Experimental Dataset 2 is built under the control of nine business activities of the loan application example process model with enacting 100 process instance event traces and storing 590 event logs (the average number of events per trace is 5.9 event logs).

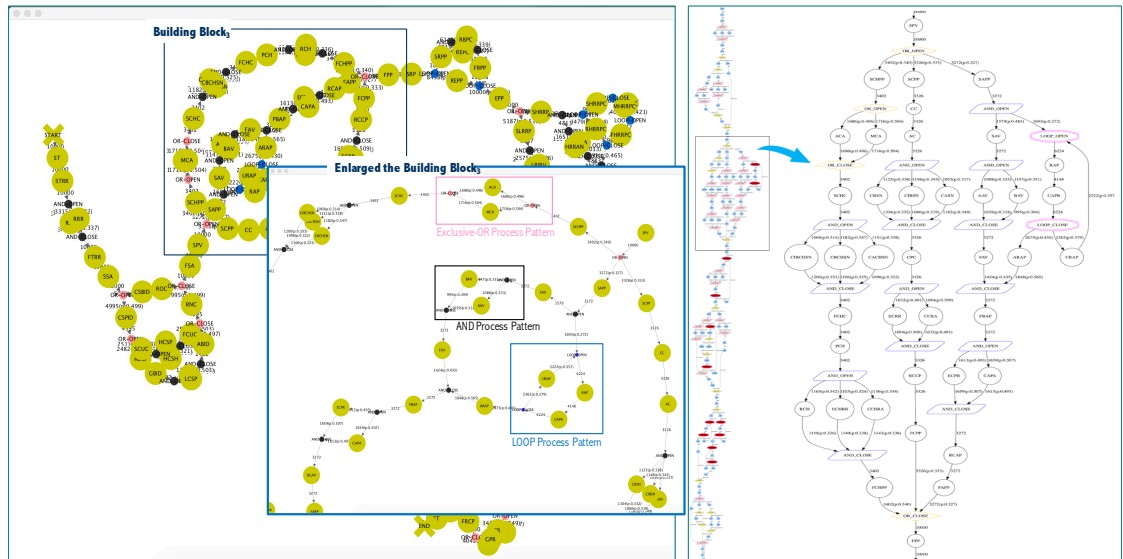

**Figure 10.** The rediscovered structured information control nets (SICN) process model from the Experimental Dataset-1.

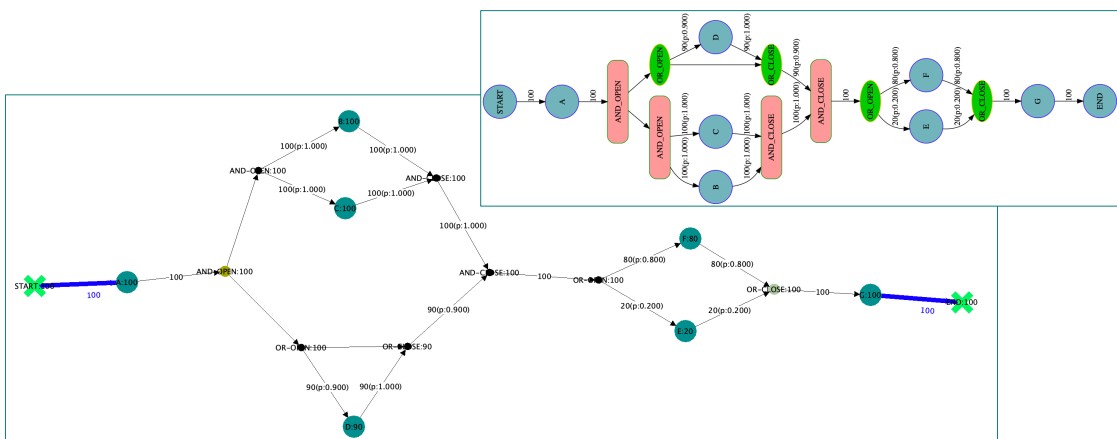

**Figure 11.** The rediscovered SICN process model with activities' occurrences from the Experimental Dataset-2.

Through these SICN process model rediscovery experiments, the process mining functionality of the integrated functional architecture and its implemented system ought to be functionally validated as you have seen. In other words, the system is able to rediscover all the structural control-flow primitive patterns, such as the conjunctive process patterns of AND-split and AND-join, the disjunctive process patterns of OR-split and OR-join, and the iterative process patterns of LOOP-split and LOOP-join, from both of the Experimental Datasets 1 and 2. Additionally, it is able to mine the number of execution occurrences on each of the 113 business activities from the Experimental Dataset 1, and the number of execution occurrences on each of the seven business activities from the Experimental Dataset 2, as well.

## 6.2. Experimental Validation I: The Process Mining with Structural Analysis

This section carries out an experimental validation study on the Experimental Dataset 1 of the large bank transaction process. Through this experimental validation study, it is proven that the implemented process mining subsystem based upon the $\rho$-Algorithm is able to completely rediscover a SICN process model and to analyze the structural aspect of the model with structural analysis statistics.

As you can see, Figure 12 shows the quantitative analysis results in terms of the number of presents on each of the structural construct types, such as conjunctive process patterns of AND-split and AND-join, the disjunctive process patterns of OR-split and OR-join, the iterative process patterns

of LOOP-split and LOOP-join, and the subprocess, and the number of execution occurrences on each of the 113 business activities on the Experimental Dataset 1. Note that these bar-chart graphs are based on the structural analysis function of the process analyzing subsystem and simply visualized by the commercialized data analysis tool.

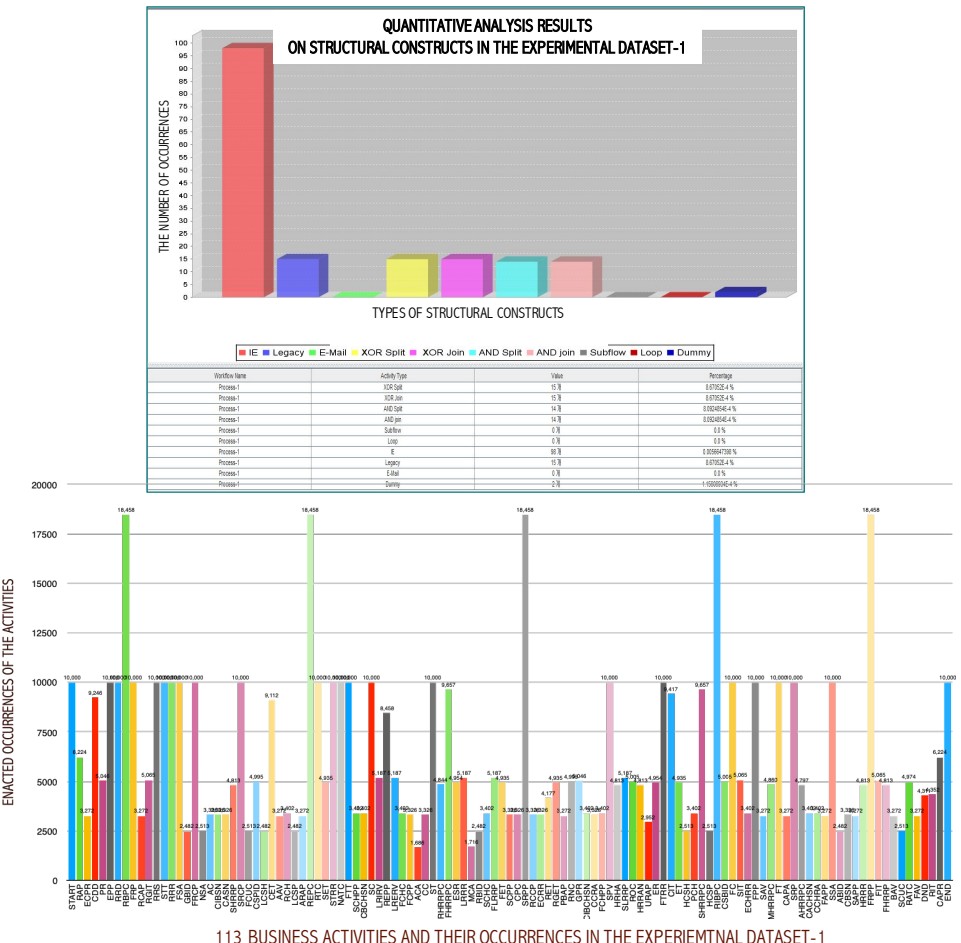

**Figure 12.** The quantitative analysis of structural constructs and business activities in the rediscovered SICN process model from the Experimental Dataset-1.

*6.3. Experimental Validation II: The Process Mining with Behavioral Analysis*

This section carries out another experimental validation study with and the Experimental Dataset 2 of the loan application example process model for evaluating the behavioral analysis functionality devised in the previous section. The behavioral analysis aspects of the implemented process analyzing subsystem are based upon the formal concept of the behavioral sequence nets that can be generated by the Algorithm 2 and the interactive and visual process instance tracing mechanism as well. First, Figure 13 shows the behavioral sequence net rediscovered from the Experimental Dataset 2 with a captured-screen and a printout form. As you can see in the figure, the process analyzing subsystem is able to rediscover every control-transition with its occurrences. Table 2 is the summary of the behavioral sequence analysis results. As you can see in the table, there are 11 behavioral sequence species rediscovered from the dataset; the behavioral sequence having the maximum number of behavioral sequences is the BS species ID, BS-7, holding 38 behavioral sequences; the average number of events per behavioral sequence is 7.73.

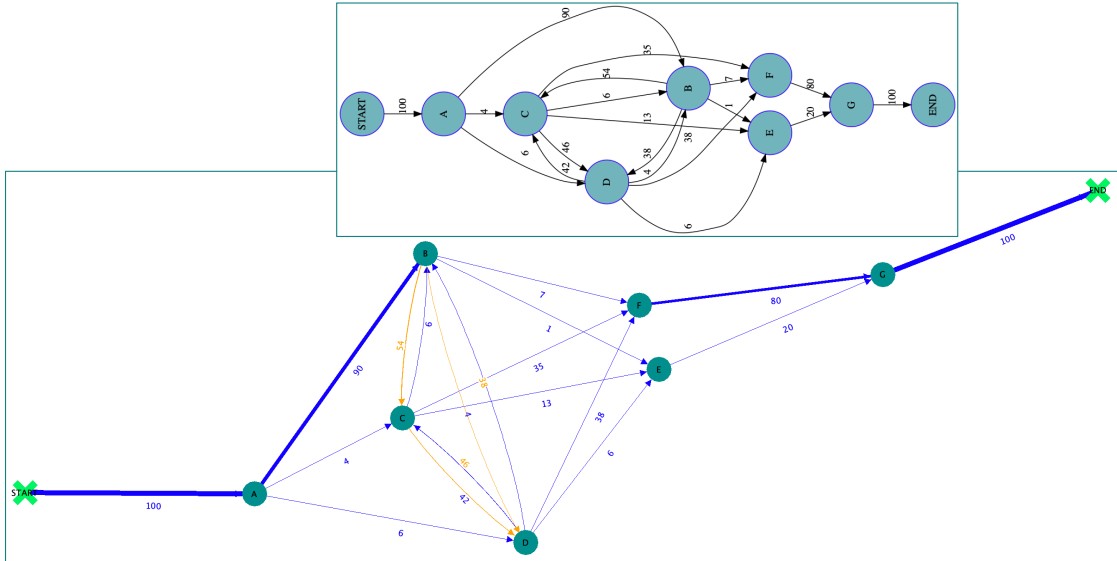

**Figure 13.** The behavioral sequence net rediscovered from the Experimental Dataset-2.

**Table 2.** The behavioral sequence (BS) analysis results on the Experimental Dataset-2.

| BS Species ID | Number of BSs | The BS IDs in the Species | Number of Events |
|---|---|---|---|
| BS-1 | 1 | 92 | 8 |
| BS-2 | 12 | 44,45,46,47,48,49,50,51,52,53,54,55 | 8 |
| BS-3 | 26 | 56,57,58,59,60,61,62,63,64,65,66,67, 68,69,70,71,72,73,74,75,76,77,78,79, 80,81 | 8 |
| BS-4 | 2 | 97,98 | 8 |
| BS-5 | 4 | 93,94,95,96 | 8 |
| BS-6 | 8 | 82,83,84,85,86,87,88,89 | 7 |
| BS-7 | 38 | 6,7,8,9,10,11,12,13,14,15,16,17,18, 19,20,21,22,23,24,25,26,27,28,29,30, 31,32,33,34,35,36,37,38,39,40,41,42,43 | 8 |
| BS-8 | 6 | 0,1,2,3,4,5 | 8 |
| BS-9 | 1 | 99 | 7 |
| BS-10 | 1 | 90 | 7 |
| BS-11 | 1 | 91 | 8 |

Based upon the Algorithm 2 and the formal concept of the behavioral sequence nets, the interactive and visual tracing functionality of behavioral sequences is implemented in the process analyzing subsystem. Therefore, the second experimental validation of the behavioral analysis aspect is carried out on the Experimental Dataset 1. Figure 14 is a captured screen of the process analyzing subsystem, on which the rediscovered SICN process model is displayed and the user is able to visually trace a behavioral sequence in an interactive fashion. In the figure, those red-colored circles represent the business activities that are already traversed during interactively running the behavioral sequence tracing session. Additionally, it supports displaying the structural statistics on the behavioral sequence traced; the behavior sequence (i.e., process instance ID, number 0) involves 70 business activities out of 113 business activities; the numbers of disjunctive exclusive-OR, conjunctive parallel-AND, and repetitive iterative-LOOP constructs involved are 7, 9 and 1, respectively.

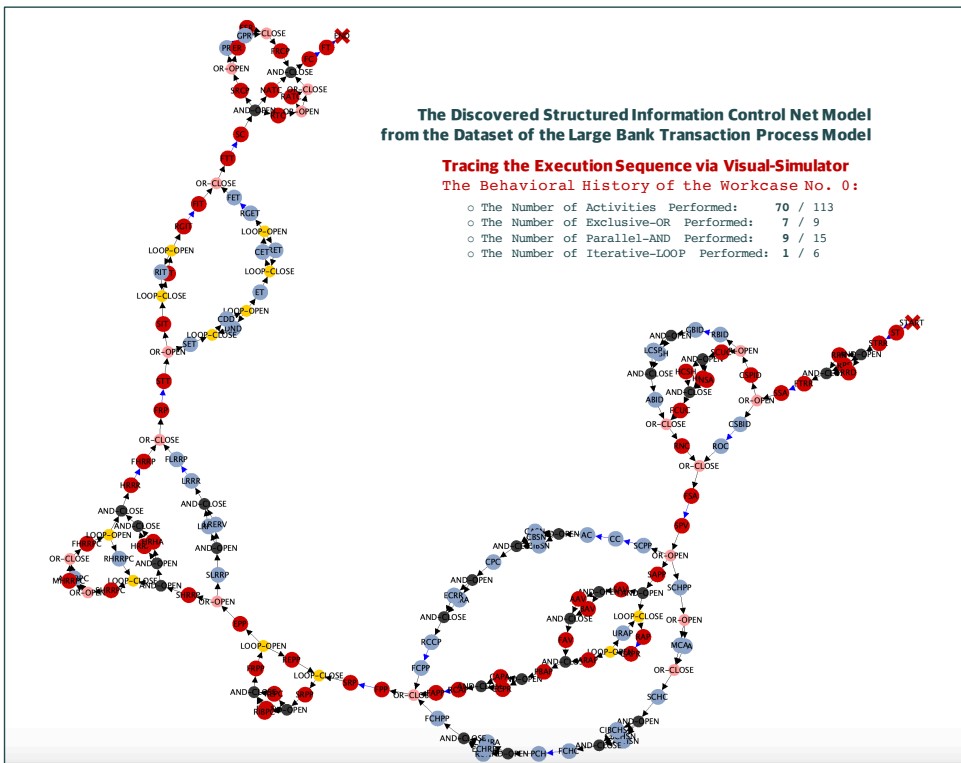

**Figure 14.** The interactive and visual tracing of behavioral sequences on the rediscovered SICN process model from the Experimental Dataset-1.

## 7. Conclusions

This paper tries to build a conceptual initiation for realizing the seamless integration of the process mining and the process analyzing functionalities. Actually, this conceptual initiation came up with a couple of the experimental validation studies dealing with a very large-scale process mining challenge. As shown in Figure 15, the rediscovered SICN process model obtained from the experimental study was made up of 624 business activities. In the experimental validation studies, it ought to be almost impossible to fulfill any further meaningful analytical work and study with the rediscovered SICN process model except the process mining work itself. This is the reason why the functional integration of the process mining and the process analyzing is issued, and this paper ought to be one of the reasonable solutions for the challenging topic in the process mining literature, even though this conceptual approach and its implemented framework will not be a complete solution for the challenging issue.

This paper has proposed the integrated functional architecture for seamlessly providing the process mining functionality and the process analyzing functionality. In order to identify the structural attributes and the behavioral process patterns in a very large-scale process model, the paper devised a series of approaches that formally extract those attributes and patterns from the structured information control nets and from the XML process definition language process models. Finally, the paper implemented the integrated functional architecture composed of a process mining subsystem and process analyzing subsystem, which are named process mining with a structural analyzer and process mining with a behavioral analyzer. Finally, the paper fulfilled a couple of experimental studies with two datasets obtained from the 4TU.Centre for Research Data and validated the feasibility of the integrated functional architecture and its implemented system. In conclusion, the issues of the process mining and analytics methodologies and systems are rapidly growing and coping with a wide diversity of application domains. So, the literature needs various, advanced, and specialized process analytical techniques and simulation methodologies in conjunction with the process mining technology,

which should be used for giving highly valuable feedback to the redesign and reengineering activities for the existing process models and packages. We strongly believe that this work might be one of those impeccable attempts and pioneering contributions for improving and advancing the process mining and analytics technology.

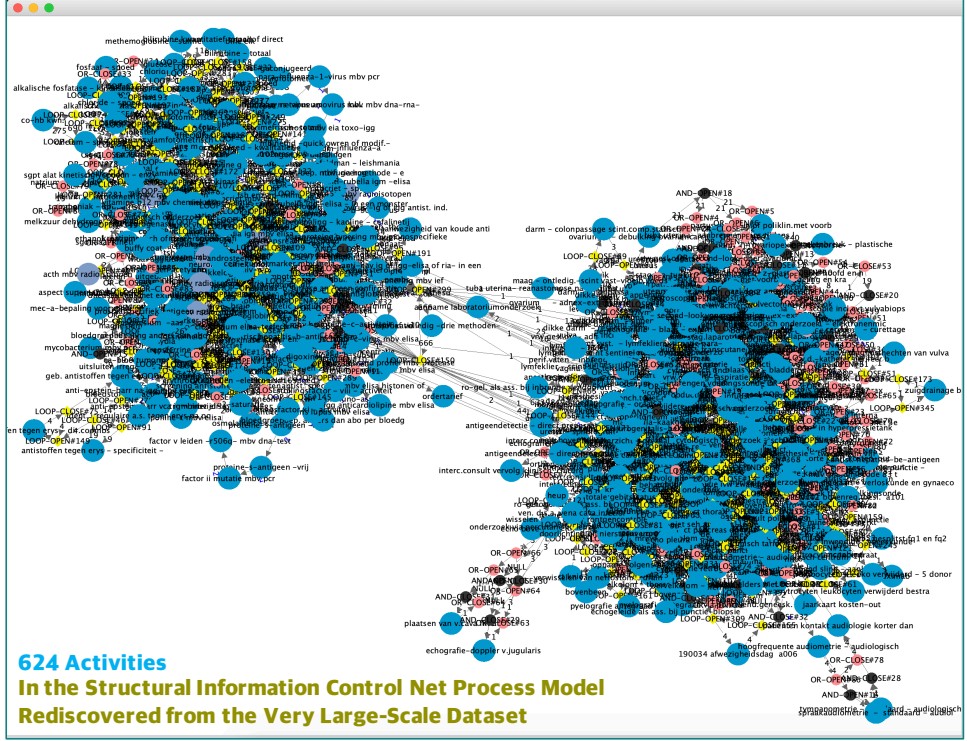

**Figure 15.** The rediscovered SICN process model with very large-scale and massively parallel business activities.

As a future work, we plan to expand the proposed integrated functionality with much more sophisticated features with the BPMN-related functionality through performing a series of experimental validation studies by using the expanded framework. Through the experimental validation studies, it ought to be expected to completely analyze the very large-scale and massively parallel process model, introduced in Figure 15, in terms of automatically analyzing the structural correctness and the behavioral properness coping with the theoretical approach with SICN and the practical approach with XPDL and BPMN. Additionally, as a more concrete future work, it is necessary for the proposed functional architecture and system to be enhanced so as for such advanced features and functions to be safely appended as the plug-in type of software components. One of the advanced features ought to be the process-aware organizational social network discovery functionality [31] that has been issued, fulfilled, produced, and implemented by the author's research group, thus far.

**Funding:** This research was supported by the KGU Research Foundation Program funded by the KYONGGI UNIVERSITY in the Republic of Korea, grant number 2017-038.

**Acknowledgments:** This research used a partial outcome of the previous research supported by Basic Science Research Program through the National Research Foundation of Korea (NRF) funded by the Ministry of Science, ICT and future Planning, Republic of Korea (Grant No. 2017R1A2B2010697). The author thanks my colleagues, and my institution for sponsoring this research. Particularly, appreciation is extended to the Kyonggi University Data & Process Engineering Research Lab student Dinh-Lam Pham; and to the 4TU.Centre for Research providing the serial IEEE XES-formatted datasets of process enactment event logs.

**Conflicts of Interest:** The author declares no conflict of interest.

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
