# Peer review of "Functional Integration with Process Mining and Process Analyzing for Structural and Behavioral Properness Validation of Processes Discovered from Event Log Datasets"

_applsci, doi:10.3390/app10041493_

Round 1
Reviewer 1 Report
The author presents an architecture which integrates Process Mining and Process Analysis, showing how a complete integration of the Mining and Analysis phases of the typical BP-life cycle could be accomplished.
They describe how going from a process model represented in the form of structured information control nets and in a graphical form of the GraphML to its logs and how to mine such logs to rediscover a process. They describe an algorithm for such mining phase which rediscover a process model in the form of XPDL process model.
The work seems solid and builds upon many previous works however I cannot recommend it for publication; for the following reasons.
Major comments:
1. The paper builds on a pile of self-referenced works. Almost half of the references are references to the author's previous works. This is nice because it describes the effort the author put in such a topic but I don't consider it fair with respect to the significant amount of work done by other colleagues which heavily studied and carried out projects over the topic of Business Process Management, BP modeling, BP analysis, BP mining, etc. To mention a few of them: Professor Wil van der Aalst and his group from Eindhoven first and now from the Aachen University; Professor Marcello la Rosa and his group from Melbourne School of Engineering; Professor Corradini and his group from University of Camerino.
2. Most of the references (which are not self-references) are outdated, they refer to publication in years: 2006, 2003, 2007. Arguments above such citations are used to justify the main contribution of this work. This is not acceptable since it means obscuring all the work done in the most recent years.
3. I don't see the need of using SICN and XPDL when we have a well established standard BPMN which has its own XML schema which makes it executable by most known engines (e.g., the Camunda engine)
4. The work focuses only on structured control-flows which have been largely studied in the literature (see La Rosa et al.). Recently high attention has been focused on providing techniques to analyze unstructured process models especially in the context of Collaboration Models (see Corradini et al.)
5. The paper presents as a contribution, a mining algorithm which has been already presented in previous works. In addition, and most disappointing, it does not provide any way to try such an algorithm no tool, no external reference to a tool or kind of, is ever provided in the entire work. This attitude undermines the reproducibility of the research and leaves doubts about the actual contribution of the presented work.
6. No comparison is provided between the proposed mining algorithm and the plethora of mining algorithm available in literature an easily accessible by means of the Process Mining tool ProM, which surprisingly is never mentioned. In addition, the time required by their approach for mining a process model is never mentioned, and no metrics for measuring the quality of the mined model is ever used/mentioned; I refer especially to metrics like fitness and precision which are highly used in literature.
7. No concrete proposal is given for future work over the presented architecture. Saying "As a future work, the paper has a plan to expand the proposed integrated functionality with much more sophisticated features" without saying which are those features is not acceptable.
Minor comments:
1.I found the work hard to read. E.g. in the introduction far to many times is repeated the term "subsystem" and complicated sentences are used along the entire text.
2. The term SICN is introduced in section 3 for the first time without explaining such acronym; the explanation comes later.
3. It is hard to believe that the paper has a plan to expand itself as stated in "As a future work, the paper has a plan to expand the proposed integrated functionality with much more sophisticated features". I guess it is the author that will expand the work done until now.
Author Response
1. The paper builds on a pile of self-referenced works. Almost half of the references are references to the author's previous works. This is nice because it describes the effort the author put in such a topic but I don't consider it fair with respect to the significant amount of work done by other colleagues which heavily studied and carried out projects over the topic of Business Process Management, BP modeling, BP analysis, BP mining, etc. To mention a few of them: Professor Wil van der Aalst and his group from Eindhoven first and now from the Aachen University; Professor Marcello la Rosa and his group from Melbourne School of Engineering; Professor Corradini and his group from University of Camerino.
--> I am recognizing and grasping those pioneers' works very well. Their theoretical background is the Petrinet-based process model. The theoretical background of this paper, however, is the ICN-based (information control net) process model that is the typical process modeling methodology proposed by my Ph. D. advisor, Professor Clarence A.(Skip) Ellis from the University of Colorado Boulder. Almost all of my research works and other colleagues' have been mainly supported by the ICN-based process model. That's the reason why this paper takes citations mainly from the author's previous works on the ICN-based process model.
2. Most of the references (which are not self-references) are outdated, they refer to publication in years: 2006, 2003, 2007. Arguments above such citations are used to justify the main contribution of this work. This is not acceptable since it means obscuring all the work done in the most recent years.
--> I revised the related works and the references by adding the recently published works in this journal.
3. I don't see the need of using SICN and XPDL when we have a well established standard BPMN which has its own XML schema which makes it executable by most known engines (e.g., the Camunda engine)
--> As mentioned before, there are typically two theoretical process models: the Petrinet and the (structured) information control net. Also, there are typically two practical and standard process definition languages: XPDL and BPML. BPMN is the graphical and notational standard. WfMC released XPDL and its extended versions with BPMN. As a vice-chair of WfMC, I have been involved in the activities of the standardized process modeling specifications.
4. The work focuses only on structured control-flows which have been largely studied in the literature (see La Rosa et al.). Recently high attention has been focused on providing techniques to analyze unstructured process models especially in the context of Collaboration Models (see Corradini et al.)
--> I agree with this comment. I would say that the proposed approach in this paper ought to be able to handle unstructured process models, too.
5. The paper presents as a contribution, a mining algorithm which has been already presented in previous works. In addition, and most disappointing, it does not provide any way to try such an algorithm no tool, no external reference to a tool or kind of, is ever provided in the entire work. This attitude undermines the reproducibility of the research and leaves doubts about the actual contribution of the presented work.
--> My research group has successfully developed a SICN-based process mining algorithm. This paper focuses on the conceptual modeling and analyzing approach and its implementable possibility. The process mining algorithm itself is not the main contribution of this paper. That's why the paper does not describe the details of the mining algorithm.
6. No comparison is provided between the proposed mining algorithm and the plethora of mining algorithm available in literature an easily accessible by means of the Process Mining tool ProM, which surprisingly is never mentioned. In addition, the time required by their approach for mining a process model is never mentioned, and no metrics for measuring the quality of the mined model is ever used/mentioned; I refer especially to metrics like fitness and precision which are highly used in literature.
--> As mentioned before and As described in the paper, the fundamental and conceptual idea came from those very large-scale process models discovered from the datasets obtained from the 4TU.Centre of Research Data and the BPI Challenges by using the process mining algorithm and system developed by my research group. Therefore, I would emphasize again that the process mining algorithm itself is not the main scope of this paper.
7. No concrete proposal is given for future work over the presented architecture. Saying "As a future work, the paper has a plan to expand the proposed integrated functionality with much more sophisticated features" without saying which are those features is not acceptable.
--> I would implicitly say that the sophisticated features ought to be related to the BPMN-related features. My research group will do this plan in the near future. I revised the sentences related to future works.
Reviewer 2 Report
After reading the paper, I have serious concerns about its appropriateness for Applied Sciences although I see some merit in the actual contents of the paper. Paper needs great revision in contents and in the writing style. In its current form, I remain skeptical of the advantages that the proposal provides.
Firstly, I recommend improving the readability and flow of the paper. In its current form, the paper is difficult for the reader to follow, and perhaps the contributions are further obfuscated because of this limitation.
Introduction section is long, but vague and weak. Introduction should show paper motivation, paper purpose and which is the paper knowledge contribution. After reading it, the research objectives and their importance are hidden. Authors should improve introduction section showing better (1) the problem that they are trying to solve, (2) the paper objectives, and (3) justifying why their proposal is necessary and its benefits.
Theoretical background/Literature review. Paper reads like an internal report instead of an academic research paper. It does not discuss alternative approaches, and it does not discuss weakness and strengths. In a scientific paper the authors have to discuss their work in context of related work and they have to elaborate what the original contribution to the state of the art is. Unfortunately, this paper fails completely in all of these aspects. The literature review is short and the results should be presented in a different way.
The case study description is very simple. Case study should be more detailed. Indeed, the research method of the case study (and the research method followed to obtain author’s proposal as well) is not well justify, so we don’t know if the results are correct or not.
On the other hand, authors should show why they have making some decisions and why they have selected some methods/tools. But nothing is explained about why these tools and their advantages
Discussion section. The contribution of the author’s approach to the literature is not highlighted. The literature review needs to be integrated with the claims that the author make in order to show the importance of its contribution. The piece is lacking in originality or a clear contribution to the literature.
I would like that authors show better the consequences for academics and practitioners of the results. It has to be showed in the conclusion section
Author Response
Firstly, I recommend improving the readability and flow of the paper. In its current form, the paper is difficult for the reader to follow, and perhaps the contributions are further obfuscated because of this limitation.
--> I reorganized the structure of the paper for improving the readability and flow of the paper.
Introduction section is long, but vague and weak. Introduction should show paper motivation, paper purpose and which is the paper knowledge contribution. After reading it, the research objectives and their importance are hidden. Authors should improve introduction section showing better (1) the problem that they are trying to solve, (2) the paper objectives, and (3) justifying why their proposal is necessary and its benefits.
--> I revised the introduction section by explicitly stating the problem and motivation, objectives, and the justification of the proposed approach.
Theoretical background/Literature review. Paper reads like an internal report instead of an academic research paper. It does not discuss alternative approaches, and it does not discuss weakness and strengths. In a scientific paper the authors have to discuss their work in context of related work and they have to elaborate what the original contribution to the state of the art is. Unfortunately, this paper fails completely in all of these aspects. The literature review is short and the results should be presented in a different way.
--> I revised the related work section by adding four recently published articles. Also, there are two theoretical process models in the literature: Petri-Net and ICN(Information Control Net). The proposed approach in this paper is based on the ICN-based process model. I would say that this paper focuses on the integrated approach of process mining functionality and process analyzing functionality for seamlessly analyzing those very large-scale process models mined from datasets of the huge number of event logs. This is the first trial in the literature. Therefore, I would focus on proposing the functionality aspect and validating the implementable possibility.
The case study description is very simple. Case study should be more detailed. Indeed, the research method of the case study (and the research method followed to obtain author’s proposal as well) is not well justify, so we don’t know if the results are correct or not.
On the other hand, authors should show why they have making some decisions and why they have selected some methods/tools. But nothing is explained about why these tools and their advantages
--> I would verify and validate the proposed approach through a tangible, visible and recognizable means, because it should be meaningful when the proposed approach is implemented. So, the mined process models rediscovered from the case studies are looked relatively simple. I would say that the datasets used in the case studies are not simple, and I would emphasize that the eventual goal of the paper is able to seamlessly analyze those huge-scale process models just like the rediscovered SICN process model shown in Figure 15.
Discussion section. The contribution of the author’s approach to the literature is not highlighted. The literature review needs to be integrated with the claims that the author make in order to show the importance of its contribution. The piece is lacking in originality or a clear contribution to the literature.
--> I revised the introduction, the related works, and the conclusion sections so as to state the highlights and the contributions as much as possible.
I would like that authors show better the consequences for academics and practitioners of the results. It has to be showed in the conclusion section
--> I revised the conclusion section.
Reviewer 3 Report
The approach developed in the paper deals with functional and behavioral models discovered thanks to process mining. The paper appears interesting and can be considered as an interesting contribution to syntactical and behavioral discovered process model research domain.
Nevertheless, the functional algorithms for extracting the structure and behavior on the discovered and parallel process models are not fully properly described, and the paper has to be improved. As far as workflow behavior is considered (for simulation and execution), several researches have been developed. Then the simulation of the workflow can be seen and discussed better within the distributed context since it is the nature of the system observed. Here the use of distributed simulation standards (such as HLA FMI/FMU) can be more discussed. As a result, I recommend reading this already articles [1] [2] [3].
Then, as an interesting perspective for the simulation of such models, some recent works expressed from event logs the time related information (e.g. state life time) and proposed discrete event based models in the frame of fuzzy [4] and [5].
The experimental study on a process enactment event log dataset available on the website of the 4TU Centre for Research Data is a good illustration of the interest of the approach.
The reference in the paper are now is too much auto centered. Please focus on main references of the author.
[1] Deelman, E., Singh, G., Su, M. H., Blythe, J., Gil, Y., Kesselman, C., ... & Laity, A. (2005). Pegasus: A framework for mapping complex scientific workflows onto distributed systems. Scientific Programming, 13(3), 219-237.
[2] Zacharewicz, G., Frydman, C., & Giambiasi, N. (2008). G-DEVS/HLA environment for distributed simulations of workflows. Simulation, 84(5), 197-213.
[3] Van Acker, B., Denil, J., Vangheluwe, H., & De Meulenaere, P. (2015, April). Generation of an optimised master algorithm for FMI co-simulation. In Proceedings of the Symposium on Theory of Modeling & Simulation: DEVS Integrative M&S Symposium (pp. 205-212). Society for Computer Simulation International.
[4] Taibi, D., & Systä, K. (2019). From Monolithic Systems to Microservices: A Decomposition Framework based on Process Mining. In 8th International Conference on Cloud Computing and Services Science, CLOSER.
[5] Wang, Y., Zacharewicz, G., Traoré, M. K., & Chen, D. (2018). An integrative approach to simulation model discovery: Combining system theory, process mining and fuzzy logic. Journal of Intelligent & Fuzzy Systems, 34(1), 477-490.
Author Response
The approach developed in the paper deals with functional and behavioral models discovered thanks to process mining. The paper appears interesting and can be considered as an interesting contribution to syntactical and behavioral discovered process model research domain.
Nevertheless, the functional algorithms for extracting the structure and behavior on the discovered and parallel process models are not fully properly described, and the paper has to be improved. As far as workflow behavior is considered (for simulation and execution), several researches have been developed. Then the simulation of the workflow can be seen and discussed better within the distributed context since it is the nature of the system observed. Here the use of distributed simulation standards (such as HLA FMI/FMU) can be more discussed. As a result, I recommend reading this already articles [1] [2] [3].
--> I revised the related works section by stating the previous works suggested by this comment. I appreciate the suggestion of the reviewer.
Then, as an interesting perspective for the simulation of such models, some recent works expressed from event logs the time related information (e.g. state life time) and proposed discrete event based models in the frame of fuzzy [4] and [5].
--> I would say that this paper mainly focuses on the conceptual integration approach and its implementable possibility. The study of [5] was proposing an integrated approach with process mining and simulation. However, the approach and goal in the paper are different from the approach and goal of mine. I appreciate the suggestion. I revised the related works section.
The experimental study on a process enactment event log dataset available on the website of the 4TU Centre for Research Data is a good illustration of the interest of the approach.
The reference in the paper are now is too much auto centered. Please focus on main references of the author.
--> I revised the references section and stated the scopes of the main references in the related works section.
Round 2
Reviewer 1 Report
The author clearly addressed most of the comments. However,
1) I still fail to grasp why all those self-citation are needed. Actually additional new self-citations that were not in the paper in the first submission have now been added. For example it is not clear which added values gives citing the following work "Ahn, H.; Pham, D.L.; Kim, K.P. An Experimental Analytics on Discovering Work Transference Networks from Workflow Enactment Event Logs. APPLIED SCIENCES 2019, 9, 2368–2391. doi:https://doi.org/10.3390/app9112368." this work actually presents a similar approach and it has some overlap with the one presented by the author; also some of the images are similar and nothing is said about the difference between the two approaches. I do understand that the paper builds upon past works of the same author but the presence for a citation it must be properly supported otherwise it makes no sense having it.
2) Still no concrete proposal is given for future work over the presented architecture. I don't think that adding just "BPMN-related features" makes it any more concrete...
Author Response
The Reviewer's Comments:
The author clearly addressed most of the comments. However,
1) I still fail to grasp why all those self-citation are needed. Actually additional new self-citations that were not in the paper in the first submission have now been added. For example it is not clear which added values gives citing the following work "Ahn, H.; Pham, D.L.; Kim, K.P. An Experimental Analytics on Discovering Work Transference Networks from Workflow Enactment Event Logs. APPLIED SCIENCES 2019, 9, 2368–2391. doi:https://doi.org/10.3390/app9112368." this work actually presents a similar approach and it has some overlap with the one presented by the author; also some of the images are similar and nothing is said about the difference between the two approaches. I do understand that the paper builds upon past works of the same author but the presence for a citation it must be properly supported otherwise it makes no sense having it.
2) Still no concrete proposal is given for future work over the presented architecture. I don't think that adding just "BPMN-related features" makes it any more concrete...
Revisions:
1) I got rid of two references of self-citation that are not directly related to the topic of the paper as the reviewer recommended. In terms of the citation that the reviewer pointed out in particular, I used the citation in the future work part of the conclusion. The reviewer asked for a concrete proposal for the future work. So, I would like to suggest a future work related to the topic that has been issued in the cited paper.
2) I proposed and stated a more concrete future work in the conclusion section.
Thank you so much for the reviewer's valuable comments!
Reviewer 2 Report
The authors have made a review of the paper considering some of my previous concerns. However, although I see some merit in their work, in my opinion paper needs revision in contents and in the writing style.
In its current form, the paper is difficult for the reader to follow, and perhaps the contributions are further obfuscated because of this limitation.
For example, which is the paper objective? See for example that authors state different paper objectives in lines 46-48, 86-87, 189-191, 474-476, 546-547.
Another example, section 5, lines 307-308 “the next consecutive sections describe the architectural and functional components of the structural analyzer and the visual simulator.” Then Section 5.1 title: “Architectural components of the Process Analyzer”
So, why Process Analyzer? It is supposed that authors are going to explain the structural analyzer. And, what has happened with the visual simulator? It has disappeared!
The same happens in other parts of the paper.
Other concerns:
- Abstract section is too long
- Introduction section mix too much information of their proposal with paper motivations. It is confused in some points
- There is no discussion section to compare their findings with previous works
In conclusion, in my opinion, this paper is not well written and it shouldn’t be published in their current form in a research journal.
Author Response
The authors have made a review of the paper considering some of my previous concerns. However, although I see some merit in their work, in my opinion paper needs revision in contents and in the writing style.
In its current form, the paper is difficult for the reader to follow, and perhaps the contributions are further obfuscated because of this limitation.
For example, which is the paper objective? See for example that authors state different paper objectives in lines 46-48, 86-87, 189-191, 474-476, 546-547.
Revision: Those lines are fully revised so as to keep the purpose of the paper consistency. I appreciated. Due to your valuable comments, the paper keeps consistency much more.
Another example, section 5, lines 307-308 “the next consecutive sections describe the architectural and functional components of the structural analyzer and the visual simulator.” Then Section 5.1 title: “Architectural components of the Process Analyzer”
So, why Process Analyzer? It is supposed that authors are going to explain the structural analyzer. And, what has happened with the visual simulator? It has disappeared!
The same happens in other parts of the paper.
Revision: Actually, the process analyzer is functionally composed of the structural analyzer and the visual simulator. So, the lines and others over the paper are completely revised so as for the readers to be not confused.
Other concerns:
- Abstract section is too long
Revision: Abstract section is reduced by getting rid of the first explaining paragraph that ought to be unnecessary as the reviewer pointed out.
- Introduction section mix too much information of their proposal with paper motivations. It is confused in some points
Revision: The introduction section is updated so as for the readers to be not confused as much as possible.
- There is no discussion section to compare their findings with previous works
Revision: In the related works section, I stated that this paper is the first trial (the proposed functional integration of SICN process mining and SICN process analyzing) in the BPM and workflow field, as I surveyed. I would emphasize that this conceptual initiation came up with a couple of the experimental validation studies dealing with a very large-scale process mining challenge like as Figure 15, where the rediscovered SICN process model obtained from the experimental study is made up of 624 business activities. I appreciated it again. Your valuable comments made the paper much clearer and more consistent.
In conclusion, in my opinion, this paper is not well written and it shouldn’t be published in their current form in a research journal.
Reviewer 3 Report
The paper has been improved and most of my comments have now been considered.
Only the validation aspect is not sufficiently detailed in the paper.
I would recommend to have a dedicated section that discusses the validation of the resulting process models.
Author Response
Reviwer's comments:
The paper has been improved and most of my comments have now been considered.
Only the validation aspect is not sufficiently detailed in the paper.
I would recommend to have a dedicated section that discusses the validation of the resulting process models.
Revisions:
In the experimental validation section, I added a dedicated subsection of "Validation of the rediscovered SICN process models," according to the reviewer's comments.
Additionally, the title of the section, "experimental studies," is changed to "experimental validation studies," and revised the related sentences allover.
Thank you so much for the valuable comments.
Round 3
Reviewer 2 Report
Paper has been significantly improved and now warrants publication